# Structural basis of Fanconi anemia pathway activation by FANCM

Rohan Bythell-Douglas [ID] [1,9], Sylvie van Twest [ID] [1,9], Lara Abbouche [ID] [1,2,9], Elyse Dunn[1], Rachel J Coulthard [ID] [3], David C Briggs [ID] [3], Vincent Murphy[1], Xinxin Zhang [ID] [1], Winnie Tan [ID] [1,4,5], Sarah S Henrikus[1], Dongming Qian [ID] [6], Yin Wu [ID] [6], Jana Wolf[7], Laurent Rigoreau[7], Shabih Shakeel [ID] [4,5], Kathryn L Chapman[7], Neil Q McDonald [ID] [3,8] & Andrew J Deans [ID] [1,2 ✉]

## Abstract

**FANCM is crucial in genome maintenance, functioning in the Fanconi anemia (FA) pathway, alternative lengthening of telomeres (ALT), and replication fork protection. FANCM recognizes branched DNA structures and promotes their remodeling through ATP-dependent branch migration. The protein has emerged as a promising therapeutic target due to synthetic lethal interactions with BRCA1, SMARCAL1, and RAD52, and in ALT-positive cancers. Here we present crystal structures of FANCM's N-terminal ATP-dependent translocase domain (2.2 Å) and C-terminal FAAP24-bound region (2.4 Å), both complexed with branched DNA. Through structural analysis, biochemical reconstitution, and cellular studies, we demonstrate that FANCM employs two distinct mechanisms: an ATP-dependent branch migration activity essential for DNA damage survival, and a branched DNA-binding mode that enhances FANCD2-FANCI monoubiquitination through FA core complex interaction. The N-terminal translocase domain specifically recognizes DNA junctions through multiple key elements, while the C-terminal FAAP24-binding domain engages adjacent double-stranded DNA. Our results reveal how FANCM evolved from an ancient DNA repair motor into a sophisticated sensor that couples DNA damage recognition to selective pathway activation, providing a structural framework for developing targeted therapeutics.**

Keywords Fanconi Anemia; Branched DNA; Translocase; DNA Damage
Subject Categories DNA Replication, Recombination & Repair; Structural Biology

## Introduction

The Fanconi anemia (FA) pathway is a critical DNA damage response mechanism responsible for the repair of replication barriers such as interstrand crosslinks (Raschle et al, 2008), R-loops (Hodson et al, 2022) and Cas9-cleavage complexes (Richardson et al, 2018). Mutations in any of the genes within this pathway lead to FA, a rare genetic disorder characterized by genome instability, bone marrow failure, developmental abnormalities and a heightened risk of cancers (Kottemann and Smogorzewska, 2013). Central to the FA pathway is the monoubiquitination of the FANCI:FANCD2 complex, which converts the protein into a clamp that coats DNA (Alcon et al, 2020; Tan et al, 2020b; Wang et al, 2021). In biochemical reconstitution experiments, FANCI:FANC2 monoubiquitination is coupled to the unloading of replication machinery (Long et al, 2014), while single-molecule studies show that FANCI:FANCD2 becomes competent for monoubiquitination when bound to dsDNA:ssDNA junctions (Alcon et al, 2024). But how activation is restricted to stalled replication forks is not yet established.

FANCM is a pivotal ATP-dependent DNA translocase protein in the FA pathway that can aid replication fork reversal (Gari et al, 2008a), and in some cases traverse (Huang et al, 2013), of DNA at replication barriers like ICLs. FANCM also functions in multiple DNA repair pathways beyond the canonical FA pathway, including by binding and recruiting Bloom syndrome proteins (Collis et al, 2008), regulating the G2/M checkpoint (Collis et al, 2008), and regulation of meiotic recombination (Crismani et al, 2012; Tsui et al, 2023). It is also a unique suppressor of Alternative Lengthening of Telomeres (ALT), and FANCM deficiency is lethal in cells using this telomerase-independent mechanism of immortality (Lu et al, 2019). FANCM's ability to bind to and remodel branched DNA structures, such as replication forks and Holliday junctions, is central to its role in all of these roles in genome maintenance (Abbouche et al, 2024a).

FANCM has emerged as a promising therapeutic target in cancer treatment, owing to its synthetic lethal interactions with various DNA

[1]Genome Stability Unit, St. Vincent's Institute of Medical Research, Fitzroy, VIC, Australia. [2]Department of Medicine (St Vincent's), University of Melbourne, Fitzroy, VIC, Australia. [3]Signalling and Structural Biology Laboratory, Francis Crick Insitute, London NW1 1AT, UK. [4]Walter & Eliza Hall Institute, Melbourne, VIC, Australia. [5]ARC Centre for Cryo-electron Microscopy of Membrane Proteins, Bio21 Institute, University of Melbourne, Parkville, VIC, Australia. [6]Viva Biotech, 201318 Shanghai, China. [7]Tessellate Bio, Stevenage, Hertfordshire SG1 2FX, UK. [8]Institute of Structural and Molecular Biology, Burbeck College School of Natural Sciences, London, UK. [9]These authors contributed equally: Rohan Bythell-Douglas, Sylvie van Twest, Lara Abbouche. ✉E-mail: adeans@svi.edu.au

repair pathways and its crucial role in ALT. Recent research has focused on targeting specific domains of FANCM, yielding promising results. For instance, disrupting FANCM's ATPase activity or MM2 domain has shown therapeutic potential in ALT-positive cancers (Lu et al, 2019; O'Rourke et al, 2019). Moreover, an ATPase-dead version of FANCM is synthetic lethal with BRCA1-deficiency and potentially more toxic than FANCM knockout (Panday et al, 2021). Finally, FANCM and its ATPase activity is also required for viability of SMARCAL1-, CIP2A, RAD52-, XPF- or WEE1-deficient cells (Aarts et al, 2015; Feng et al, 2024; Wang et al, 2018).

Another primary function of FANCM is in the monoubiquiti-nation of FANCD2 and FANCI in the FA pathway. Both mice and humans with FANCM deficiency show significantly impaired, almost absent FANCD2 monoubiquitination (Bakker et al, 2009; Bogliolo et al, 2018; Tsui et al, 2023). Because FANCM contains two DNA-binding domains, it was proposed at its discovery to be a potential anchor and activator of the FA core complex (Abbouche et al, 2024a; Niedernhofer, 2007). Important for this function was its central MM1 domain, which binds the FA core complex component FANCF (Deans and West, 2009). In this way, FANCM directly links binding of damaged DNA to the ubiquitination machinery of the FA core complex.

Recent structural studies of the FA core complex have revealed the intricate architecture of this large (1MDa), asymmetric complex. It contains nine proteins including two copies of the FANCL E3 RING ligase positioned at opposite ends of a FANCB-FAAP100 dimer (Shakeel et al, 2019; Wang et al, 2021). One of these FANCL enzymes interacts with a tetramer of FANCC, FANCE, and FANCF. FANCE from this subcomplex binds DNA-associated FANCD2 and brings it into proximity of the second FANCL subunit, to catalyze ubiquitin transfer from the E2 conjugating enzyme UBE2T (Wang et al, 2021). Once the channel between the C-terminal domains of FANCD2 and its partner FANCI close around the bound DNA, ubiquitin attached to FANCD2 acts as a molecular pin, thereby trapping the dsDNA inside(Alcon et al, 2020; Tan et al, 2020b; Wang et al, 2021). Subsequent monoubiquitination of FANCI further stabilizes this clamped conformation and protects the ubiquitin on FANCD2 from deubiquitination by the USP1-UAF1 enzyme complex (Rennie et al, 2021; Rennie et al, 2020; van Twest et al, 2017).

Here, we investigate the structural biology of FANCM, focusing on its ATPase and DNA-binding domains as well as FANCM interactions with the FA core complex. We identify the key features responsible for FANCM's ATP-dependent branch migration activity but surprisingly, we demonstrate that this activity is dispensable for FANCI and FANCD2 monoubiquitination in the FA pathway. Nevertheless, mutants deficient in either ATPase or FA core complex binding remain sensitive to replication-blocking DNA damage, highlighting at least two separable roles of FANCM in the maintenance of genome stability.

# Results

To understand how FANCM coordinates its multiple functions in genome maintenance, we first investigated its biochemical activity in the FA pathway (Fig. 1A). We purified the complete FANCM anchor complex (FANCMc), containing FANCM and its partner proteins FAAP24, MHF1, and MHF2 (Figs. 1A,B and EV1A).

When added to reconstituted FA core complex reactions, FANCMc dramatically enhanced FANCD2 and FANCI mono-ubiquitination (Fig. 1C,D). This stimulation showed striking DNA structure specificity-branched DNA substrates that mimic damaged replication forks were much more effective than simple double-stranded DNA (Fig. 1C). Most notably, while FANCD2 modification increased about 2.5-fold, FANCI monoubiquitination was enhanced over 20-fold, reaching substantially higher maximum levels (Fig. 1D). The strongest stimulation was observed with poly-dIdC DNA, which forms web-like structures(Pichlmair et al, 2009) mimicking complex DNA damage sites.

This preferential enhancement of FANCI modification has important implications for pathway regulation. The deubiquitinat-ing enzyme USP1:UAF1, a therapeutic target in cancer (Rennie et al, 2024), normally removes ubiquitin from FANCD2 unless FANCI is also modified (Lemonidis et al, 2023; Rennie et al, 2020; van Twest et al, 2017). In the presence of FA core complex and USP1:UAF1, the slow conversion of FANCD2$^{Ub}$-FANCI to FANCD2$^{Ub}$-FANCI$^{Ub}$ means that USP1:UAF1 deubiquitination normally dominates (van Twest et al, 2017). However, we found that FANCMc enables robust dual modification even at high USP1:UAF1 concentrations (Fig. 1E,F), suggesting it promotes stable pathway activation by rapidly driving FANCI ubiquitination to protect modified FANCD2.

Supporting the physiological relevance of this activity, FANCMc overcame the salt sensitivity of the FA core complex-mediated reaction. While FANCD2:FANCI monoubiquitination was impaired even at sub-physiological salt concentrations, FANCMc maintained robust activity under physiological conditions (Fig. EV1B), indicating it plays a direct role in stabilizing the reaction.

## Distinct DNA-binding domains coordinate FANCM's role in pathway activation

To understand how FANCM recognizes DNA damage sites, we systematically analyzed its multiple DNA-binding domains. FANCM forms a complex with three partner proteins (MHF1, MHF2, and FAAP24), each contributing potential DNA-binding surfaces (Abbouche et al, 2024a). Through a series of "dropout" experiments removing different components, we made the surprising discovery that MHF1/2—previously shown to enhance FANCM's DNA motor activity (Yan et al, 2010)—were dispensable for stimulating FANCD2 and FANCI monoubiquitination (Fig. 1G, lanes 12–14).

In striking contrast, removal of FAAP24, which binds FANCM's C-terminal domain (Ciccia et al, 2007), severely compromised the stimulatory activity (Fig. 1G, lanes 9–11). Equally critical was the N-terminal Hel2i domain (residues 298–433), known to recognize branched DNA structures (Abbouche et al, 2024b). Deletion of this region not only abolished stimulation but also actively inhibited the monoubiquitination reaction (Fig. 1H). Moreover, the isolated FANCM translocase domain (residues 82–647) showed no stimulatory activity (Fig. 1G, lanes 6–8), indicating that both N- and C-terminal DNA-binding modules are essential.

These results reveal that FANCM employs a sophisticated dual recognition mechanism, using distinct DNA-binding surfaces to identify damage sites and activate the FA pathway. This bipartite recognition system could explain how FANCM restricts pathway activation to specific DNA structures while maintaining its separate motor functions.

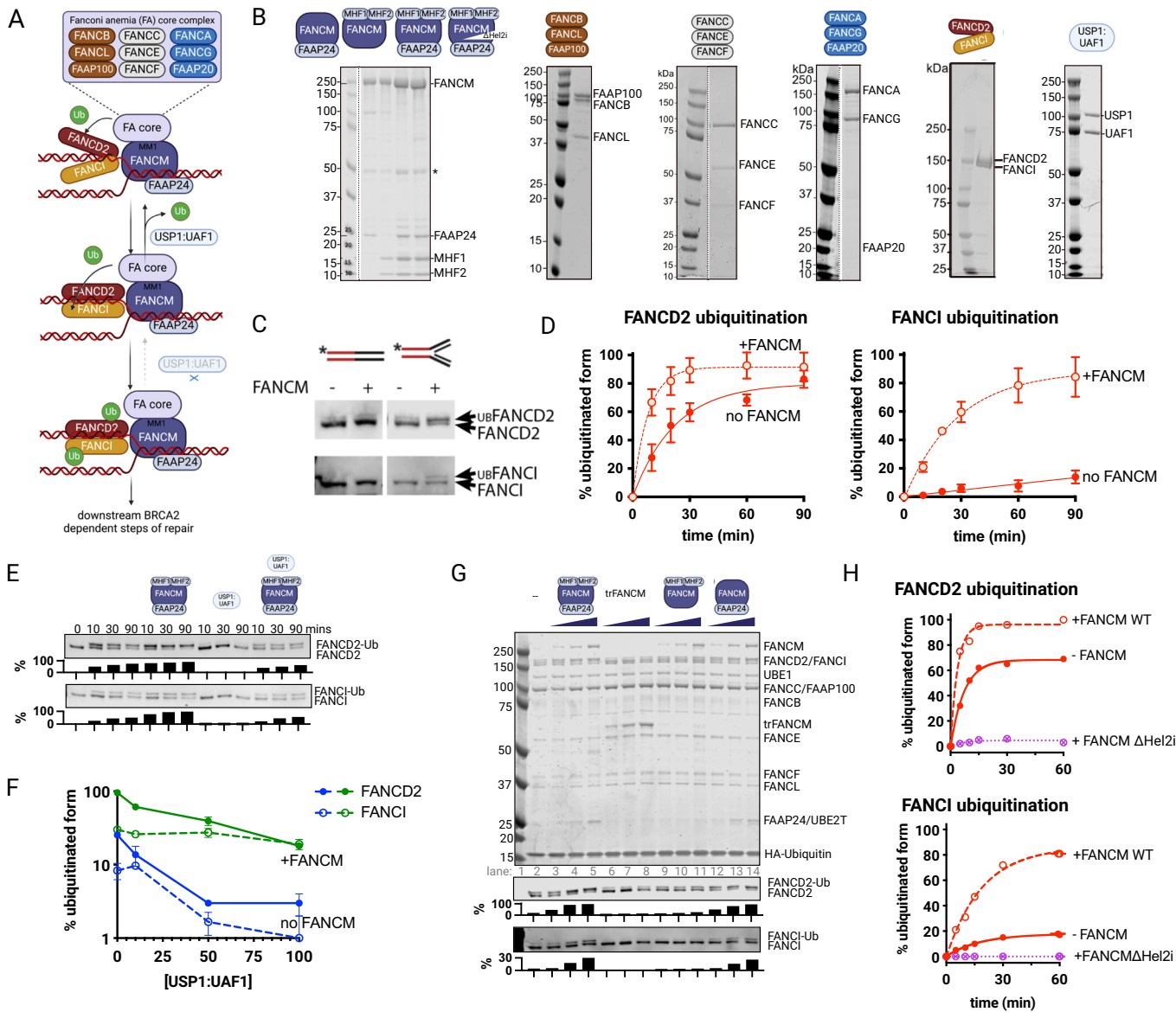

**Figure 1. FANCM stimulates the activation of the Fanconi anemia pathway by promoting FANCD2 and FANCI monoubiquitination.**

(A) Schematic of the FA pathway and role of FANCM. Step 1: FANCI:FANCD2 recognises ssDNA:dsDNA junction, while FANCM complex recognises fork structure and recruits FA core complex to monoubiquitinate FANCD2, step 2: FANCD2-Ub:FANCI is clamped on DNA, FANCI is monoubiquitinated OR USP1:UAF1 deubiquitinates FANCD2, step 3: FANCD2-Ub:FANCI-Ub becomes permanent clamp to activate FA pathway, step 4: downstream BRCA2-dependent steps of fork repair. (B) Coomassie blue-stained proteins used in this study. (C) FANCMc stimulation of FANCD2:FANCI monoubiquitination in the presence of fork but not linear DNA. (D) Time-course experiment of FANCD2 (left) or FANCI (right) monoubiquitination by FA core complex with or without FANCM. Values are average +/−SD from three independent experiments. (E) Example Western blots of FANCD2 and FANCI monoubiquitination by FA core complex in the presence of 100 nM USP1:UAF1 or 100 nM FANCMc. Quantification shows percentage FANCD2 or FANCI monoubiquitinated. (F) FANCD2 and FANCI monoubiquitination in the presence of USP1:UAF1, quantification shows mean +/−SD of three experiments competition experiment shows that FANCMc promotes accumulation of FANCD2-Ub:FANCI-Ub at equal ratios in the presence of increasing concentrations of USP1:UAF1. (G) Example Coomassie blue-stained PAGE gel of ubiquitination reactions and Western blots of FANCD2 or FANCI when using 10, 30, or 100 nM FANCMc, FANCM 72–684 (trFANCM), FANCM:MHF1:MHF2 or FANCM:FAAP24, Quantification shows percentage FANCD2 or FANCI monoubiquitinated. (H) Quantification of time-course experiments of FANCD2 (top) or FANCI (bottom) monoubiquitination in the presence of FANCMc WT, FANCMc ΔHel2i, or FANCMcΔFAAP24. Source data are available online for this figure.

## Structural basis for branched DNA recognition by FANCM

To understand how FANCM identifies DNA damage sites, we determined the crystal structure of its N-terminal translocase domain bound to branched DNA at 2.2 Å resolution (Fig. 2A,B). This high-resolution view reveals a sophisticated molecular machine built around three core elements: two RecA-like domains (RecA1 and RecA2) that form the motor unit, connected by a specialized insert domain (Hel2i) that recognizes DNA branch points (Fig. 2C). A unique helical extension wraps around these elements, creating a precisely shaped groove that captures branched DNA.

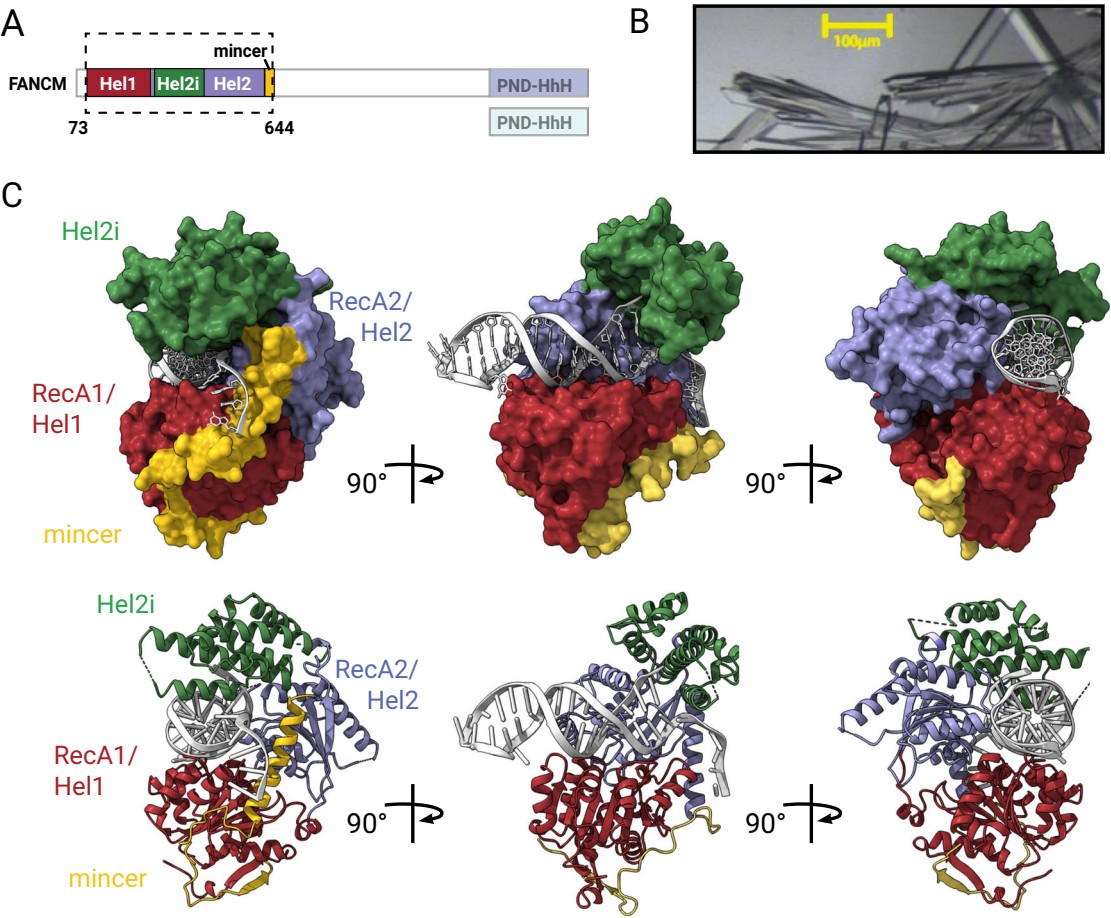

**Figure 2. Crystal structure of the FANCM translocase domain bound to DNA.**

(A) Domain organization of FANCM and the fragment used for crystallization (dotted box). (B) Example crystals and (C) structure of translocase domain bound to 3'Flap hairpin DNA. Structure is shown from three orientations, in surface (top) and cartoon (bottom) representation, with subdomains colored as in (A), and DNA in silver.

The overall architecture shows both conserved and unique features compared to related SF2 family machines. The RecA1-RecA2 interface is dominated by polar interactions, and key structural elements that stabilize this interface are concentrated around residues C110-L115 and A246-G249 in the RecA1 domain and E549-D551 and I568-576 in RecA2. This architectural arrangement mirrors the RecA1-RecA2 domain organization found in numerous SF2 family proteins, including MDA-5, LGP2, RIG-I, the DNA nuclease XPF and the siRNA enzyme Dicer (Fig. EV2A). Although the individual RecA1, RecA2, and Hel2i domains structurally align closely with their counterparts in these homologs (Fig. EV2B), the relative position of these subdomains to each other within the translocase domain differs notably, even when compared to structures in similar nucleotide-free states (Fig. EV2C). These distinct conformations likely serve two purposes: the variable positioning between RecA1 and RecA2 is linked to the inherent requirement of flexibility for ATPase activity, while the different Hel2i domain orientations may determine substrate specificity. In particular, we observed that the opening angle between RecA2 and Hel2i is markedly reduced in FANCM compared to the other enzymes, all of which process double-stranded nucleic acids. This tighter configuration is likely key to the binding of the branchpoint by FANCM, as compared to a looser configuration,

which permits the passage of double-stranded molecules over the RecA1-RecA2 interface in structurally related proteins.

Perhaps most striking is FANCM translocase domain's divergent C-terminal region. In RIG-I a "pincer" domain at this position is characterized by two long α-helices centered on a rigid 60° bend that folds back from RecA2 across RecA1 (Fig. EV2D). While the N-terminal α-helix maintains a comparable trajectory in FANCM, the C-terminal helix adopts a distinct orientation, positioned at ~130° relative to the first helix and traversing an alternative molecular surface. We observed that deletion of residues 590–640 encompassing this region, led to the absence of stable protein production (Fig. EV2E). Because of this modified and distinct pincer architecture, we renamed this sequence as the "mincer" domain. The relative contributions of each FANCM domain to DNA binding will be discussed later.

## FANCM integrates multiple DNA-binding domains to recognize damage sites

To understand how FANCM's different regions cooperate in DNA recognition, we further determined the crystal structure of its C-terminal region bound to FAAP24 and branched DNA at 2.4 Å resolution (Fig. 3A–C; Appendix Table S1). Two copies of the

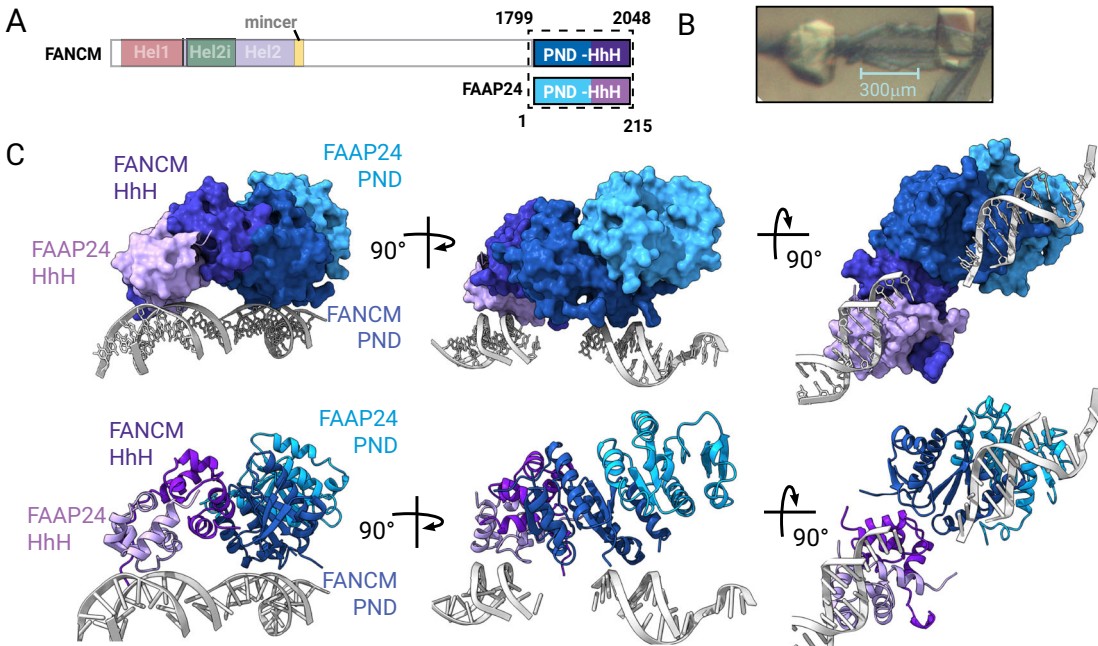

**Figure 3. Crystal structure of FAAP24:FANCM-CTD bound to splayed DNA reveals duplex binding by FANCM ERCC4 domain and FAAP24 HhH domains.**

(A) Domain organization of FANCM and the fragment used for crystallization (dotted box). (B) Example crystals and (C) structure shown from three orientations in cartoon (top) or surface (bottom) representations, with subdomains colored as in (A), and DNA in silver.

complex are present in the asymmetric unit, related by non-crystallographic symmetry, which bind the duplexes in a head-to-tail arrangement (Fig. EV3A). The splayed arm DNA bases were not visible in the electron density, indicating that they were conformationally disordered and therefore not bound specifically by the protein (Fig. 3C).

A previous FANCM^ERCC4:FAAP24 structure by our team (PDB:4BXO) was determined under high calcium conditions (Coulthard et al, 2013) from protein crystallized with only linear DNA. In contrast to that structure which showed calcium ions at each FAAP24 (HhH)₂ domain-DNA interface and in the FANCM pseudonuclease domain (PND) active site, our new structure showed magnesium ions in the first FAAP24 HhH domain and another in the FAAP24-PND outside the active site (Fig. EV3B).

The new structure also reveals a previously unappreciated 5 bp dsDNA binding site within the FANCM pseudonuclease domain complementing the known contact with an additional five base pairs of DNA by FAAP24's HhH domain (Fig. 3C). Separated by an eight basepair gap, this arrangement enables recognition of approximately 18 continuous base pairs, substantially more than previously known DNA-binding sites in the complex (Fig. EV3A,B).

To understand how these DNA-binding modules coordinate their activities, we used AlphaFold3 modeling combined with our crystal structures. The models predict that FANCM's N- and C-terminal domains contact the same DNA molecule but grip opposite strands (Fig. 4A,B; Appendix Fig. S1A–C). Two key interfaces connect these domains: N-terminal residues S199/R200/K234 contact C-terminal regions around I1827/V1847, while the Hel2i domain interacts with a conserved loop (residues 1900-1914). While I1827/V1847 mediate crystal contacts, solution studies

confirm these regions participate in domain interactions rather than oligomerization (Fig. EV3A,D,E).

Mutation of the highly conserved interface residues I1827 and V1847 confirmed their functional importance. Though these mutations didn't affect DNA binding by the isolated C-terminal domain (Appendix Fig. S1D), they severely compromised FANCD2 foci formation and cellular resistance to mitomycin C (Fig. 4C,D). A patient-derived mutation truncating the C-terminal HhH domain (p.Arg1931*; rs144567652) associated with cancer predisposition (Peterlongo et al, 2015) and infertility (Kasak et al, 2018) showed similar defects, and is consistent with previous observations that both the C-terminal region and FAAP24 are essential for FANCM function (Coulthard et al, 2013; Yang et al, 2013).

These findings suggest a sophisticated DNA damage recognition system, where the FANCM motor domain identifies branch points while the C-terminal domains provide additional grip and positioning.

## FAAP24 coordinates DNA binding with efficient ATP-driven branch migration

FANCM substrates include recombination intermediates (Gari et al, 2008b; Tsui et al, 2023) and replication barriers such as interstrand crosslinks (Raschle et al, 2008), R-loops (Hodson et al, 2022) and Cas9-cleavage complexes (Richardson et al, 2018). The discovery that FANCM's domains physically interact suggested they might functionally cooperate in migrating these branched DNA structures. To test this, we developed a novel real-time assay using fluorescent DNA junctions that report branch migration through separation of dye-quencher pairs (Fig. 5A). This system revealed that FANCM's activity is strictly ATP-dependent, as

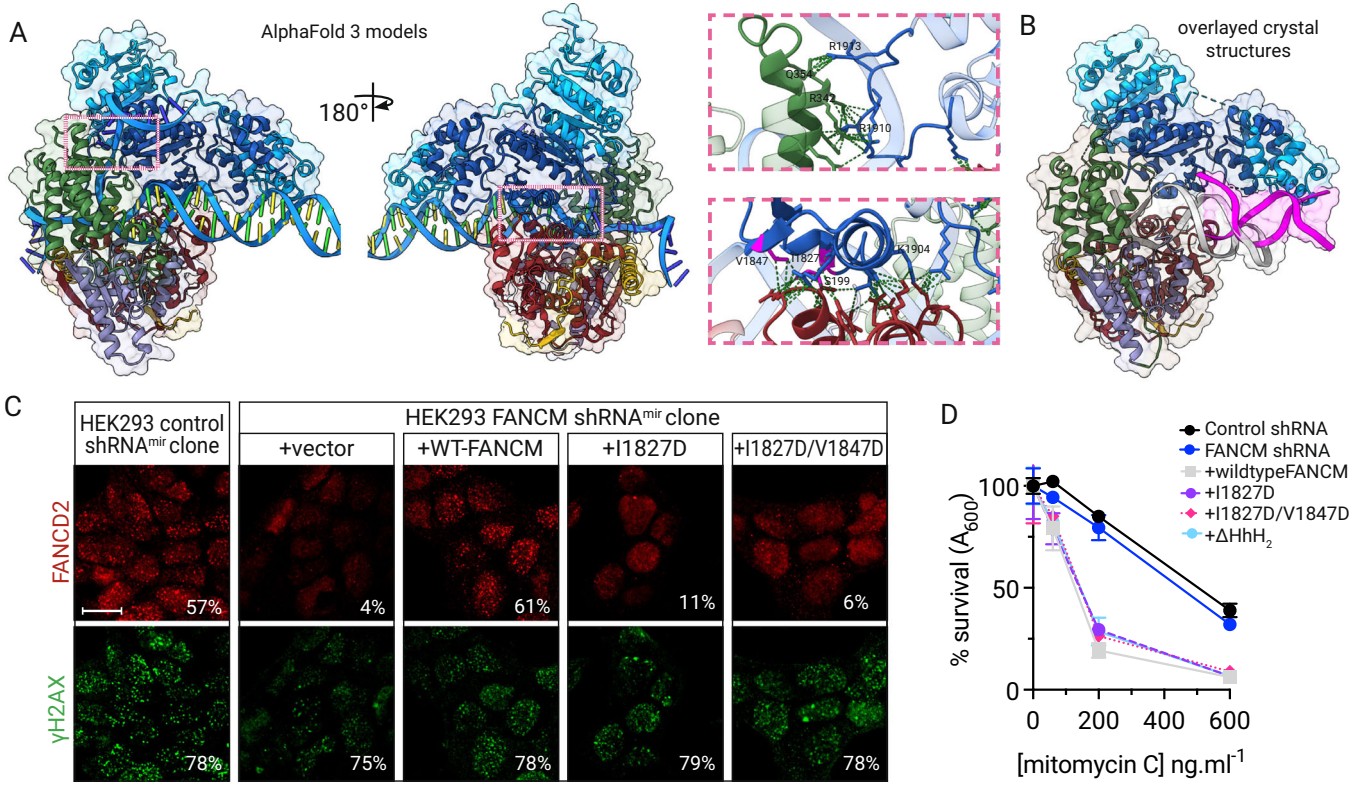

**Figure 4.  FANCM stimulates the activation of the Fanconi anemia pathway by promoting FANCD2 and FANCI monoubiquitination.**

(A) Cartoon and transparent surface model of FANCM:FAAP24 generated using AlphaFold3 as per methods. Pink boxes show points of contact between N- and C-termini, which are shown inset at higher magnification. (B) Actual crystal structures aligned to the same orientation as domains in (A). DNA of translocase domain co-crystal shown in silver, DNA from FANCM-CTD:FAAP24 co-crystal shown in pink. (C) FANCD2 and γH2AX foci formation in HEK293 control shRNA[mir] or FANCM shRNA[mir] clones complemented with vector, FANCM, or indicated FANCM mutants, after 7 h of 40 ng/ml mitomycin c exposure. Inset numbers represent percentage of cells with >5 bright foci, out a total of 200 scored nuclei. Scale bar = 10 μm. (D) Survival as measured by percentage SRB staining at 96 h post treatment with indicated dose of mitomycin c. Points represent the mean of six independent replicates +/− SE. Source data are available online for this figure.

non-hydrolyzable ATP analogs abolished branch migration (Fig. 5B). Importantly, branched DNA stimulated ATPase activity 5–7-fold more than linear DNA, confirming structure-specific activation (Fig. 5C).

By simultaneously monitoring branch migration together with ATP consumption, we made the surprising discovery that while the isolated translocase domain consumes higher levels of ATP than FANCM complex, its activity is threefold lower (Fig. 5D). An inefficient coupling of ATPase and translocase activity was also seen when FAAP24 was removed from the complex (Fig. 5E), revealing that FANCM's C-terminal domains optimize its motor efficiency.

### Structure-specific DNA recognition drives efficient branch migration

Our crystal structures revealed how FANCM achieves this efficiency through precise DNA contacts. First, the N-terminal domain creates a positively charged channel that grips DNA at the branchpoint (Figs. 5F and EV4). Two key residues in the Hel2i domain—H369 and Y332—make specific contacts with the 5′ and 3′ sides of the junction respectively (Fig. 5F, inset 1). Mutations of either residue impair both ATP hydrolysis and branch migration (Fig. 5G), confirming their role in motor activation.

The newly identified mincer domain provides additional structure-specific recognition through a long helix that contacts the 3′ DNA arm (Fig. 5F, inset 2). Remarkably, mutations in this helix (Q600F or K610F) specifically reduce branch migration while preserving ATPase activity (Fig. 5H). This domain appears to be a unique adaptation, as related enzymes like RIG-I and Dicer lack equivalent nucleic acid contacts (Fig. EV2), suggesting it evolved specifically to couple ATP hydrolysis to migration of branched DNA structures.

### FANCM has distinct roles in pathway activation and DNA processing

A crucial question in FANCM biology has been whether its motor activity is required for FA pathway activation. Using ATP-γ-S (which blocks ATP hydrolysis at the γ-phosphate bond, but not β-phosphate cleavage by E1 enzymes, Fig. 6A), we made the surprising discovery that while this completely abolished branch migration (Fig. 5B), it had no effect on FANCM's ability to stimulate FANCD2 and FANCI ubiquitination (Fig. 6B).

To confirm this unexpected separation of functions, we tested three "ATPase-dead" mutations (K117R, D214A, and V555F) targeting different motifs of FANCM's motor domain. While these

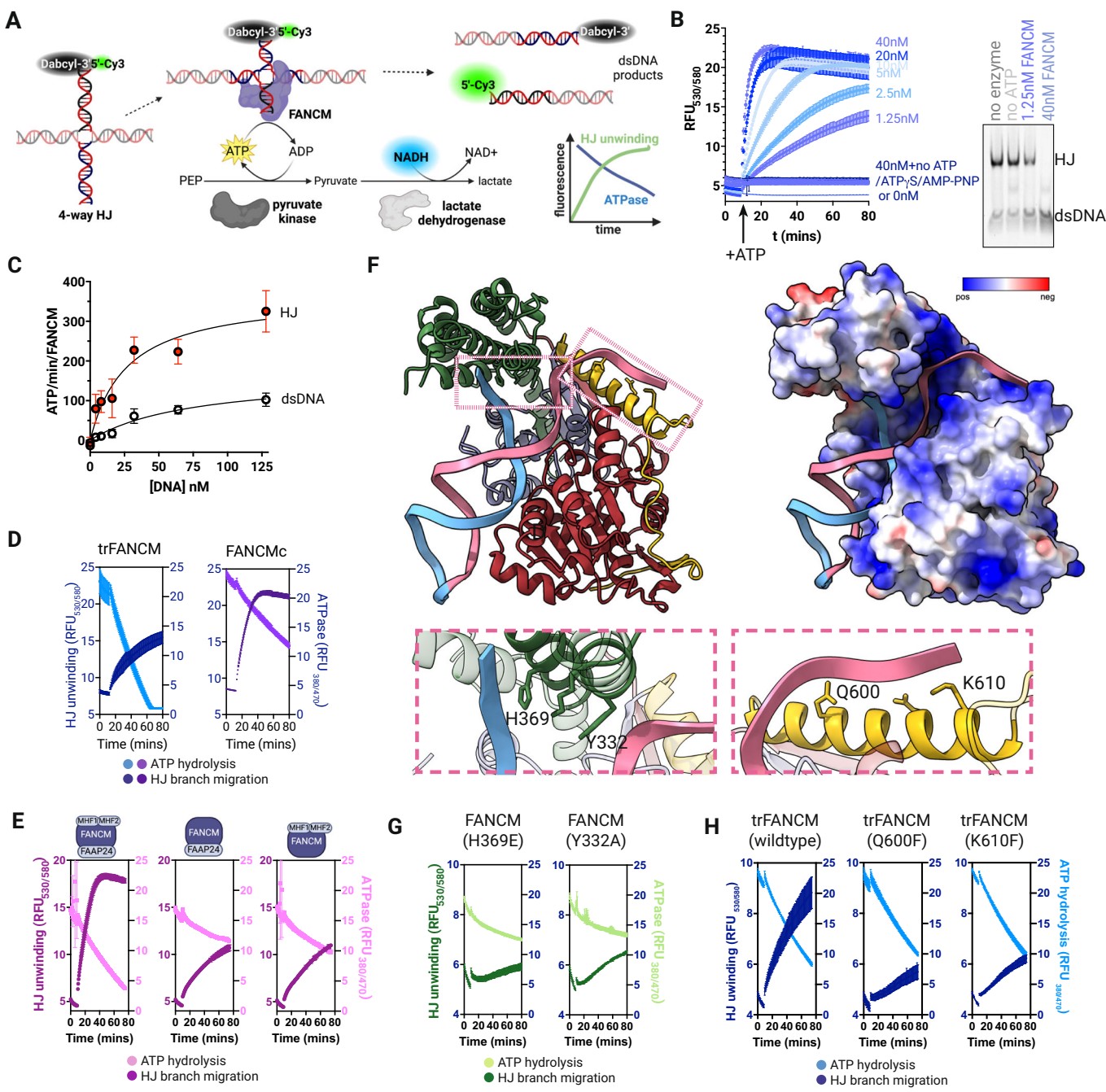

**Figure 5. Key DNA-binding residues in the translocase domain are required for coupling ATPase and branch migration activity.**

(A) Schematic of real-time branch migration assay and coupling with ATPase assay. (B) Example real-time assay using different concentrations of FANCMc vs max concentration with no ATP or with ATPγS or AMP-PNP. (C) Stimulation of ATPase activity of the FANCM translocase domain by increasing the concentration of HJ-DNA or dsDNA. (D) Coupling experiment showing that unwinding rate is similar, but ATP hydrolysis is higher for the translocase domain. (E) Coupling experiment for full-length FANCM with no FAAP24 or no MHF present. (F) Structure of DNA-binding regions of FANCM translocase shown as cartoon with domains colored as in Fig. 2, or as surface showing charge (Coulombic coloring as per legend). Insets show close-up of regions within pink boxes: critical amino acids at the junction (inset 1) or on the mincer domain (inset 2). (G) Coupled DNA translocase and ATPase assays are both reduced with FANCMc-H369E and Y332A and (H), but only the translocase activity is affected in trFANCM mincer domain mutants K600F or K610F. In all graphs, mean ± SD is shown from $n = 3$ experiments. Source data are available online for this figure.

mutations eliminated both ATPase activity and branch migration (Fig. 6C), they maintained normal stimulation of FANCD2 and FANCI monoubiquitination (Fig. 6D). This revealed that FANCM's motor activity is dispensable for canonical FA pathway activation through FANCD2:FANCI.

Y332 and H369 in the N-terminal translocase domain contact the 3′ and 5′ sides of the junction, respectively, and mutations to either equally inhibit FANCM's ATPase and translocase activity (Fig. 5G). Surprisingly, these mutants differ in their ability to stimulate FANCD2 and FANCI monoubiquitination. The H369E

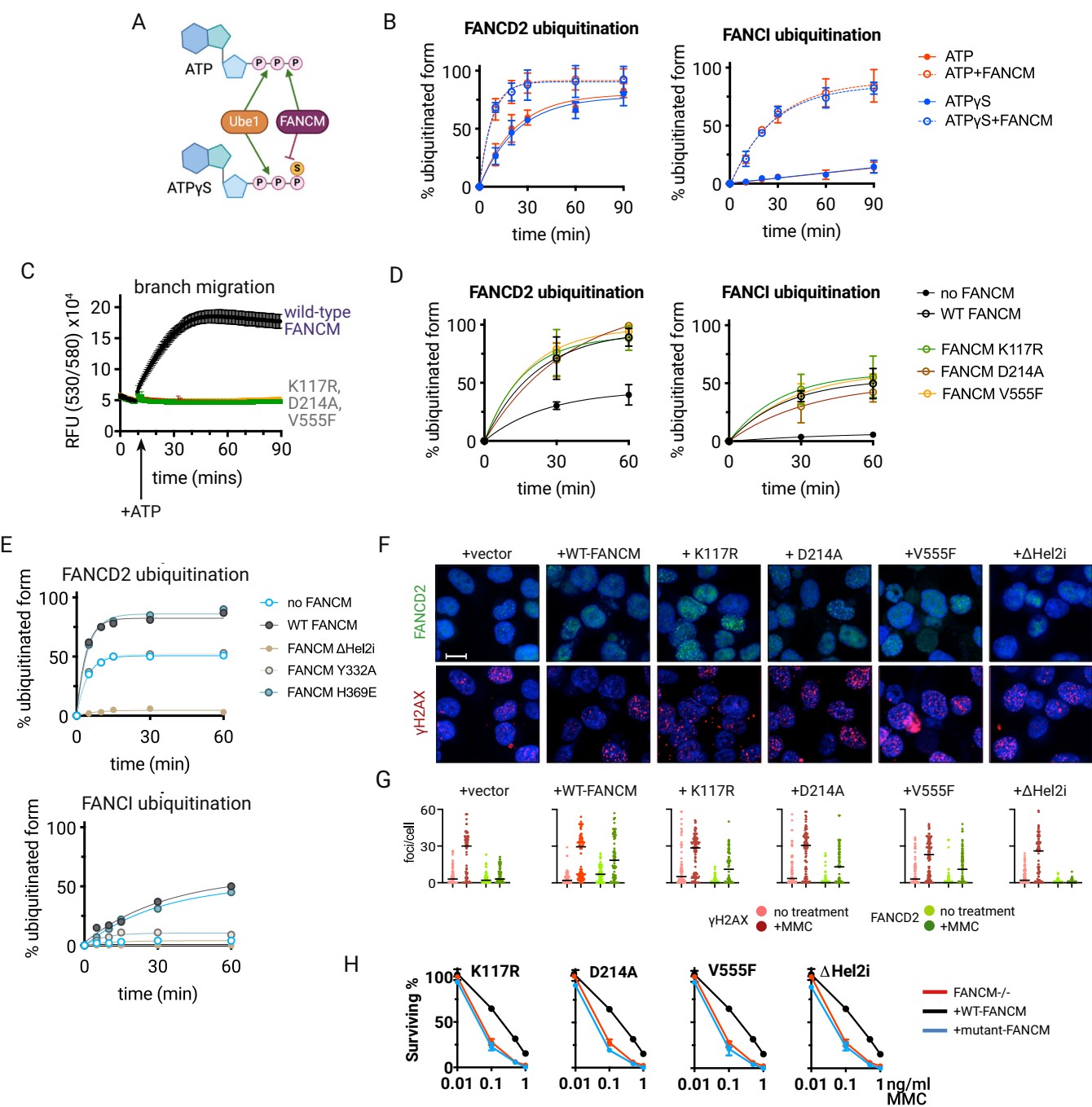

**Figure 6. FANCM structure-specific binding, but not ATPase activity, is required for FANCD2:FANCI monoubiquitination.**

(A) Schematic showing how FANCM and UBE1 differently utilize ATP and ATPγS. (B) Time course of FANCD2 and FANCI monoubiquitination by FA core complex is invariant with FANCM (open circles) or no FANCM (filled circles) in either the presence of ATP (red) or ATPγS (blue). Values shown are average ± SD of three independent experiments. (C) Branch migration activity of wild-type FANCM, or FANCM-K117R, D214A or V555F ATP hydrolysis defective mutants. Values shown are average ± SD of three independent experiments. (D) Time course of wild-type or ATP hydrolysis mutant FANCM-stimulated FANCD2 and FANCI monoubiquitination by FA core complex. Note: Different rates due to different enzyme concentration used compared to 6B. Values shown are average ± SD of three independent experiments. (E) As in (A), but with indicated structure-specific DNA-binding mutants. (F) FANCD2 (green) and γH2AX (red) foci formation in HCT116 FANCM−/− cells complemented with: empty vector, WT FANCM, K117R, D214A, V555F, ΔHel2i after treatment with 10 ng/ml mitomycin C for 7 h. Scale bar = 10 μm. (G) Quantification (foci counts/nucleus) of cells shown in (F), 100 nuclei scored per condition. (H) MMC sensitivity of HCT116 FANCM−/− after transduction with vector (red line), wild-type FANCM (black line) or indicated FANCM mutants (blue line). Graphs represent mean ± SD of three independent experiments. Source data are available online for this figure.

mutation behaves like other ATPase-defective mutants, maintaining normal levels of FANCD2:FANCI activation (Fig. 6E). In contrast, the Y332A mutation completely abolishes the stimulatory effect of FANCMc on FANCD2 and FANCI monoubiquitination (Fig. 6E). While the reason for this difference isn't immediately clear, it suggests that proper engagement of either the 3′ flap or the branchpoint base (both of which interact with Y332) is crucial for stimulating FANCD2 and FANCI monoubiquitination.

In cellular studies, the purely ATPase-defective FANCM-K117R, -D214A, and -V555F mutants maintained normal levels of mitomycin C-activated FANCD2 foci formation, while FANCM-ΔHel2i did not (Fig. 6F,G). However, supporting the conclusion that both ATPase activity and FANCD2 monoubiquitination are required for cell survival after crosslinker damage, all four mutants were equally sensitive to mitomycin C (Fig. 6H). Thus, altogether these findings demonstrate that FANCM's ATPase activity contributes differently to core complex stimulation and its other DNA damage response-related functions.

### Patient mutations reveal how FANCM recruits the FA core complex

Recent structural studies of the FA core complex have revealed the intricate architecture of this large (1MDa), asymmetric complex. It contains nine proteins including two copies of the FANCL E3 RING ligase positioned at opposite ends of a FANCB-FAAP100 dimer (Shakeel et al, 2019; Swuec et al, 2017; Wang et al, 2021). One of these FANCL enzymes interacts with a tetramer of FANCC, FANCE and FANCF. FANCE from this subcomplex binds DNA-associated FANCD2 and brings it into proximity of the second FANCL subunit, to catalyze ubiquitin transfer from the E2 conjugating enzyme UBE2T (Wang et al, 2021), but FANCF was shown to be non essential (van Twest et al, 2017). Through a combination of predictive modeling (using "Predictomes" (Lim et al, 2023)), and biochemical analysis, we identified FANCF as FANCM's direct binding partner in the FA core complex (Fig. 7A). Supporting this, we found that FANCM specifically co-purifies with a FANCC:FANCE:FANCF subcomplex (Fig. 7B). While not being required for basal FA core complex activity (Fig. 7C, lanes 2–3), we discovered FANCF is absolutely required for FANCM-mediated stimulation of FANCD2 and FANCI monoubiquitination (Fig. 7C, lanes 5–6).

AlphaFold modeling revealed the structural basis for this interaction. FANCM residues 984–1021 form two conserved helices that bind across FANCF's N-terminal domain (Fig. 7D,E). This region, known as the MM1 domain, adopts a distinctive structure with three conserved prolines creating a 40° kink. The binding site on FANCF remains accessible within the full FA core complex (Appendix Fig. S2), and is adjacent to residues 943–1021 which encompass a phosphodegron region (Kee et al, 2009). An additional helix in FANCM (residues 959–968) is also predicted to contact FANCC, potentially stabilizing the interaction.

The significance of this interface became clear when we examined cancer-associated FANCM variants. Mutations found in Australian women with a strong history of breast and ovarian cancer but not *BRCA1* or *BRCA2* mutation (Li et al, 2021) are located in the MM1 domain, and appear essential for FANCF binding. Two such variants, V990A and P999L, completely abolished interaction with the FA core complex, while E964K in the FANCC-binding region reduced association by 60% (Fig. 7I).

These mutations prevented both FANCD2:FANCI ubiquitination in vitro (Fig. 7F) and DNA damage responses in cells (Fig. 7G,H,J).

These findings reveal how FANCM activates the FA pathway: after recognizing branched DNA through its motor domains, it uses the MM1 domain to recruit the FA core complex through FANCF. This mechanism explains both how pathway activation is restricted to DNA damage sites and why specific FANCM mutations can cause cancer predisposition.

## Discussion

In summary, our structural and biochemical analyses reveal how FANCM evolved to serve dual roles in genome maintenance through distinct mechanisms. While its ancient DNA branch migration activity is shared with archaeal (HEF) and yeast (Mph1/Fml1) homologs (Hodson et al, 2022; Komori et al, 2004; Nishino et al, 2005; Prakash et al, 2009), FANCM acquired a sophisticated DNA damage recognition system in vertebrates. The key to this evolution was the MM1 domain, which co-opted FANCM into the FA pathway by acquiring the ability to couple its DNA-binding activity to FA core complex recruitment, while maintaining its ancestral branch migration function (Fig. EV5A).

This mechanistic separation has important implications for disease and therapy. FANCM's dramatic enhancement of FANCI monoubiquitination (>20-fold) ensures rapid conversion of the initial FANCD2-Ub clamp into a stable, USP1-resistant FANCD2-Ub:FANCI-Ub complex (Fig. EV5B) (Rennie et al, 2020; van Twest et al, 2017). This is likely the main mechanism by which FANCM promotes activation of the canonical FA pathway through FANCD2 monoubiquitination. And it achieves this through co-opting the DNA-binding properties of the protein. The N-terminal translocase specifically recognizes branched DNA through its Hel2i and mincer domains, while the C-terminal region provides additional grip through FAAP24-mediated DNA binding. Notably, MHF1/2 are required only for ATPase functions, not for FA core complex stimulation. However, MHF1-knockout cells show reduced FANCD2 monoubiquitination, probably because FANCM is degraded in their absence. While MHF1/2 are essential for ICL traverse (Abbouche et al, 2024a), future studies with our separation-of-function mutants could determine whether this traversal activity (Huang et al, 2013) also depends on FANCM's ATPase function, DNA binding, or both.

The distinct biochemical signatures of patient mutations - ATPase-dead variants causing infertility versus MM1 mutations in breast cancer - reflect FANCM's separable functions. For example, the V555F variant, which causes male infertility through Sertoli only syndrome (Zhang et al, 2021), completely lacks ATPase activity while retaining normal FANCD2:FANCI monoubiquitination stimulation. This contrasts with MM1 domain mutations identified in non-BRCA1/2 breast cancer families (Li et al, 2021), which show the opposite pattern-disrupted FANCD2:FANCI monoubiquitination but intact ATPase activity. The separable nature of FANCM's ATPase and FA pathway activation functions suggests that a comprehensive analysis of patient variants, coupled with their biochemical defects, could reveal genotype-phenotype correlations that inform both disease prognosis and therapeutic strategies. Such correlations would be particularly valuable given FANCM's emerging role as a cancer predisposition gene (Figlioli et al, 2023; Peterlongo et al, 2021) and its

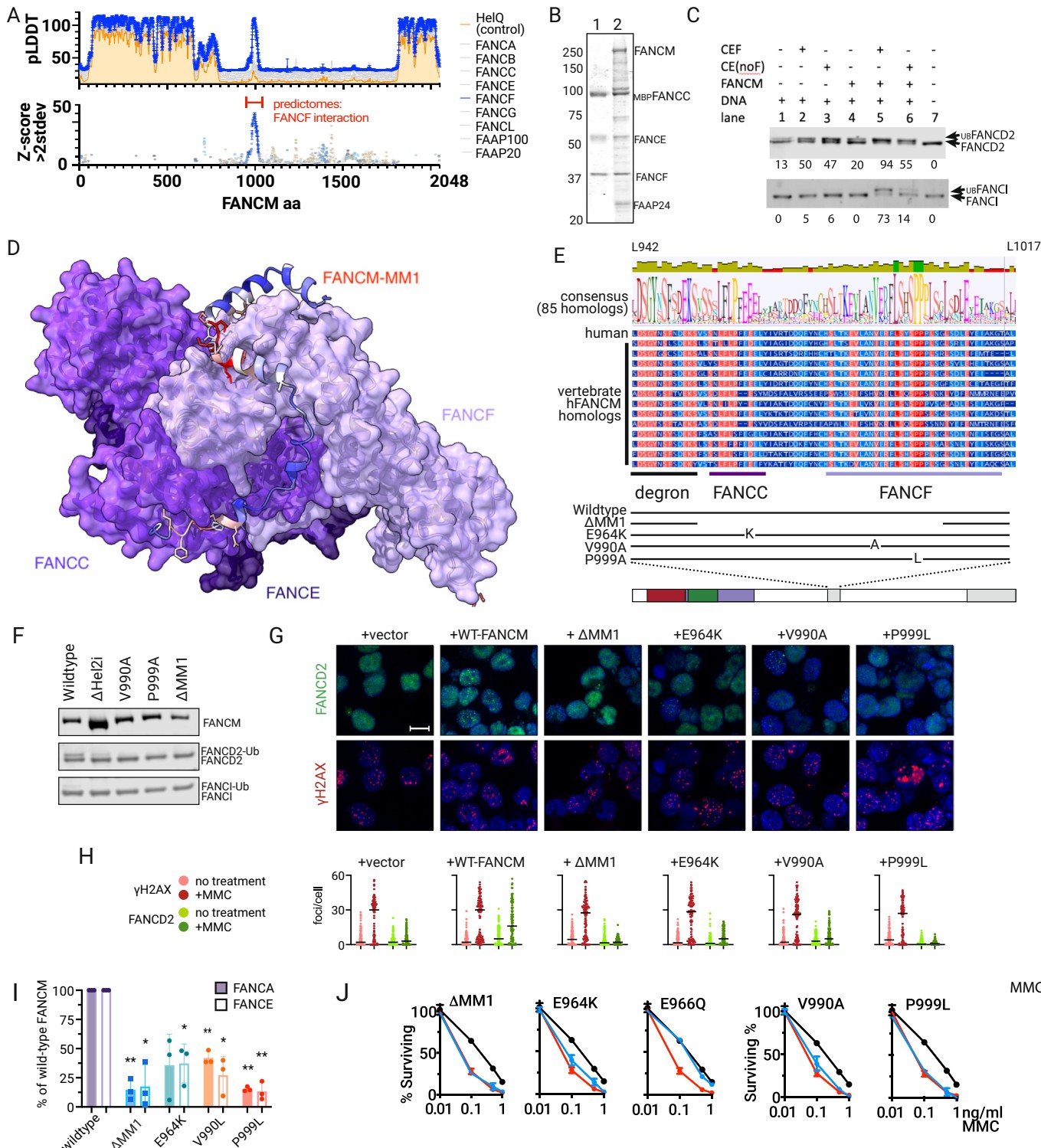

synthetic lethal interactions with multiple DNA repair pathways (Panday et al, 2021).

The separable biochemical activities of FANCM also provide new opportunities for developing selective inhibitors. While complete loss of FANCM is lethal in ALT+ cells (Lu et al, 2019;

O'Rourke et al, 2019; Pan et al, 2019; Silva et al, 2019), our structural and biochemical analyses suggest that specifically targeting the ATPase activity might achieve better therapeutic outcomes than complete protein inhibition. Our high-resolution structures of the FANCM translocase domain bound to DNA reveal

◀ **Figure 7. FANCM:FANCF interaction promotes the activation of FANCD2 and FANCI monoubiquitination in the presence of branched DNA.**

(A) Predicted local distance difference test (pLDDT) at amino acids 1–2048 of FANCM when co-folded at predictomes.org with each of the FA core complex components, compared to a control. Each plot is mean ± SD ($n = 3$ models) offset by 2% to highlight similarities in all maps except that with FANCF. Values > 2 SD from mean plotted below, with the predicted FANCF interacting region from these maps. (B) Amylose copurification of FANCM:FAAP24 with MBP-FANCCEF. Lane 1: MBP-FANCC-E-F only, Lane 2: MBP-FANCE-E-F + FANCM:FAAP24. (C) Stimulation of FANCD2 and FANCI monoubiquitination requires FANCF. (D) AlphaFold model of FANCM bound to FANCC:FANCE:FANCF complex. Coloring of MM1 domain indicates conserved residues as per the alignment shown in (E) which also indicates location of mutants tested. (F) FANCM-ΔMM1, E364K, V990A and P999L mutants do not stimulate FANCD2:FANCI monoubiquitination in in vitro assays. (G) Example FANCD2 (top, green) and γH2AX foci (bottom, red) in FANCM−/− cells complemented with empty vector, Flag-FANCM-wildtype, ΔMM1, E364K V990A or P999L mutant proteins. Scale bar = 10 μm). Quantification of results shown in (H), from at least 100 nuclei/condition. (I) Reduced association of Flag-FANCM-ΔMM1, E364K V990A and P999L with FA core complex components FANCA or FANCE in cell-based immunoprecipitation experiments (results shown are quantified Western blots of FANCE and FANCA from $n = 3$ independent co-immunoprecipitation experiments) *$P < 0.05$, **$P < 0.01$, one-way ANOVA with Dunnett's multiple comparison test, exact $P$ values for each comparison provided in source data. (J) MMC sensitivity of HCT116 FANCM−/− after transduction with vector (red line), wild-type FANCM (black line) or indicated FANCM mutants (blue line). Graphs represent mean ± SD of three independent experiments.a Source data are available online for this figure.

unique features of the DNA-binding sites, and how they couple branched DNA binding to translocation, that could be exploited for selective inhibitor design. Such compounds would create a dominant-negative form of FANCM that retains DNA binding and core complex recruitment while blocking its ability to process DNA structures—precisely the combination that previous studies suggested would be most toxic to cancer cells (Lu et al, 2019; Panday et al, 2021; Silva et al, 2019).

## Methods

### Reagents and tools table

| Reagent/resource | Reference or source | Identifier or catalog number |
|---|---|---|
| **Experimental models** | | |
| HEK293-FLPin cells | Thermo Fisher | R75007 |
| HEK293-FLPin-FANCM[shRNAmir] | Deans and West, 2009 | |
| HCT116 FANCM-/- | Wang et al, 2013 | |
| **Recombinant DNA** | | |
| pADC10 | Hodson et al, 2022 | |
| pADC10-6HIS-FANCM_73-644 | This study | |
| pGEX-GST-FANCM(1799-2048)-HIS-FAAP24 | Coulthard et al, 2013 | |
| pOG44 | Thermo Fisher | V600520 |
| pDEST-FRT-3xFlag-FANCM | Deans and West, 2009 | |
| pDEST-FRT-3xFlag-FANCM variants | This study | |
| FANCM targeted shRNA[mir] sequence | Open Biosystems | V2HS_203688 |
| **Antibodies** | | |
| Anti-FLAG M2 antibody resin | Sigma | A2220 |
| Anti-Flag antibody (M2) | Sigma | F1804 |
| Anti-StrepII antibody | Abcam | ab76949 |
| Anti-FANCE | Bethyl | A302-125A |
| Anti-FANCA | Bethyl | A301-980A |
| Anti-FANCM (CV5.1) | Vuono et al, 2016 | |
| Anti-FAAP24 (SWE94) | Ciccia et al, 2007 | |
| Anti-FANCD2 | Novus | NB100-182 |

| Reagent/resource | Reference or source | Identifier or catalog number |
|---|---|---|
| Anti-γH2AX | Upstate | 16-193 |
| **Oligonucleotides and other sequence-based reagents** | | |
| Oligos for migration assays | This study | Appendix Table S2 |
| **Chemicals, enzymes, and other reagents** | | |
| Sf900 Serum Free media | Thermo Fisher | |
| Mammalian protease inhibitor cocktail | APExBIO | K1007 |
| FLAG peptide | Sigma | F3290 |
| MgCl$_2$ | | |
| Adenosine 5'-Triphosphate (ATP) | New England Biolabs | P0756L |
| PreScission protease | Tan et al, 2020a | |
| Glutathione sepharose | | |
| His-UBE1 | Boston Biochem | E304 |
| FANCC-FANCE-FANCF complex | van Twest et al, 2017 | |
| FANCA-FANCG-FAAP20 complex | van Twest et al, 2017 | |
| FANCB-FANCL-FAAP100 complex | van Twest et al, 2017 | |
| Avi-tag ubiquitin | Tan et al, 2020a | |
| HA-ubiquitin | Boston Biochem | U-110 |
| FANCM-FAAP24-MHF1-MHF2 complexes | This study | "Methods" |
| Phospho(enol)pyruvic acid tri(cyclo-hexylammonium) salt | Sigma | P7252-500MG |
| Pyruvate kinase | Roche | 10128155001 |
| L-lactate dehydrogenase | Roche | 10127230001 |
| Poly dIdC | Sigma | P4929 |
| IPTG | | |
| ATPγS | Sigma | A1388 |
| AMP-PNP | Sigma | A2647 |
| SYBR Gold | Thermo Fisher | |
| Hygromycin B | Thermo Fisher | |
| Mitomycin C | Selleck Chem | S8146 |

| Reagent/resource | Reference or source | Identifier or catalog number |
|---|---|---|
| **Software** | | |
| GraphPad Prism 10 | GraphPad | |
| REFMAC5 | Murshudov et al, 2011 | |
| DUI2 | Winter et al, 2018 | |
| AlphaFold3 | www.alphafoldserver.com | |
| **Other** | | |

## Protein purification

Full-length FANCM was purified as previously described (Hodson et al, 2022). The genetic construct for expressing the N-terminally FLAG-tagged FANCM translocase domain (residues 82–647), or mutant derivatives, was synthesized with a codon bias for expression in insect cells in a transposition-compatible vector pADC10 (Hodson et al, 2022). Plasmids were transformed into DH10Multibac for integration into the MultiBac genome (Berger et al, 2004). Mutant forms were generated by in vitro mutagenesis or gene synthesis (Gene Universal), and expressed and purified using the same methods as wild-type protein. trFANCM and variants were expressed in Hi5 insect cells at 26 °C for 72 h post infection in Sf900 Serum Free media. Cells were harvested by centrifugation at $1000 \times g$ for 15 min). All purification steps were conducted at 4 °C. Cells were lysed by sonication in buffer (50 mM HEPES pH 7.0, 300 mM NaCl, 1 mM TCEP, 10% v/v glycerol, 1× mammalian protease inhibitor cocktail (APExBIO, K1007)). The lysate was clarified by centrifugation at $15,000 \times g$ for 30 min. Clarified lysate was applied to pre-equilibrated anti-FLAG M2 antibody resin (0.5–1 ml compact resin) and incubated with gentle rolling at 4° for 1 h. The resin was collected via centrifugation at $1000 \times g$ for 15 min at 4 °C, then washed 3× in the same manner with ~30 ml purification buffer (50 mM HEPES pH 7.0, 300 mM NaCl, 1 mM TCEP, 10% v/v glycerol). Resin was transferred to a gravity flow column, and an ATP wash (1 mM ATP, 2 mM MgCl₂ in purification buffer) was performed. The ATP was washed away with 10× CV purification buffer, and the protein was eluted with 4× CV of 0.2 mg/ml flag peptide in purification buffer.

For crystallization of the FANCM NTD, a slightly different construct 8HIS-FANCM$_{73-644}$ was used, with purification by Nickel affinity purification and elution with imidazole, followed by heparin and size exclusion column purification.

FANCM-CTD-FAAP24 was purified as previously described (Coulthard et al, 2013). Briefly, pGEX-GST-FANCM$_{(1799-2048)}$-HIS-FAAP24 was transformed into BL21-Rosetta-pLysS (Novagen). Single colonies were cultured until OD$_{600}$ = 0.4. followed by induction with 25 μM IPTG and incubated overnight at 18 °C, followed by cell pelleting at $4000 \times g$ for 20 min. Cell pellets. After pelleting, cells were resuspended in 20 ml ice-cold extraction buffer (500 mM NaCl, 50 mM Tris pH 7, 1 mM DTT, 5% glycerol, 0.1% BOG, 10 mM benzamidine, 1 mM AEBSF) per 1 L culture and sonicated (4 × 30 s bursts, 45 s rest on ice), then centrifuged at $29,200 \times g$ for 30 min at 4 °C. The supernatant was incubated with glutathione sepharose for 60 min, followed by PreScission protease cleavage in buffer P (100 mM NaCl, 20 mM Tris pH 7.5, 1 mM DTT, 5% glycerol). Peak fractions were combined and loaded onto

HiTrap HP heparin columns, washed in buffer P, and eluted using salt gradients (100 mM to 1 M NaCl); fractions were analyzed by SDS-PAGE before final purification by size exclusion chromatography using 30-cm superdex 200 column. Peak fractions were pooled and concentrated.

Avi-Ubiquitin, UBE2T, Flag-FANCI, StrepII-FANCD2, Flag-FANCA-FANCG-FAAP20 and MBP-FANCC-FANCE-FANCF were purified as previously described (Tan et al, 2020a; van Twest et al, 2017). His-UBE1 was purchased from Boston Biochem.

## Protein analysis

A nanodrop ND-1000 spectrophotometer was used to measure protein concentrations (mg protein/ml) via the absorption at 280 nm, with the use of theoretical molar extinction coefficients from the Expasy ProtParam server (http://www.expasy.org/tools/protparam/).

Dynamic light scattering measurements were performed using a Zetasizer Nano-S (Malvern). Protein samples were concentrated to >0.5 mg/ml, 0.22 μm filtered, and analyzed using standard instrument protocols.

Fluorescence anisotropy measurements used 5′-6-carboxyfluorescein (6FAM)-labeled oligonucleotides (Sigma Genosys UK). Oligonucleotides were dissolved to 100 μM in TE buffer (10 mM Tris.HCl pH 7.5, 1 mM EDTA, 50 mM NaCl) and annealed 1:1 with complementary strands by heating to 95 °C followed by slow cooling. Measurements were performed in triplicate using a TECAN-SAFIRE2 fluorometer (λex = 490 nm, λem = 520 nm) in 20 μl reactions containing 0.1 mg/ml BSA, 25 mM Tris.HCl pH 8, 2 mM MnCl₂, and 20 mM NaCl using freshly gel-filtered protein and 384-well black low-volume non-binding microplates (Corning).

## FANCD2:FANCI monoubiquitination experiment

Standard FA core complex monoubiquitination reactions were carried out at 37 °C in a total volume of 20 μL. Each reaction included 10 mM recombinant Avi-ubiquitin, 20 nM human recombinant UBE1, 150 nM UBE2T, 100 nM FANCA:FANCC:FANCE, 150 nM FANCB:FAAP100-FANCL (wildtype or mutant), 100 nM poly dIdC (Sigma), and 100 nM FANCI:FANCD2 complex and 2 mM ATP. In Fig. 1C, 60 bp dsDNA or replication fork structures were used in place of poly dIdC, purified as in (Abbouche et al, 2024b). Reactions were stopped using NuPage LDS sample buffer and 5 min at 80 °C. Reaction products were detected by Western blotting with anti-Flag antibody (M2, Sigma, 1:1000) or anti-StrepII antibody (Abcam ab76949). Electromobility shift assays were performed as in Tan et al (Tan et al, 2020b).

## Generation of nucleic acid substrates

All nucleic acid substrates were generated by annealing synthesized ssDNA oligonucleotides (IDT) as per Appendix Table S2.

To generate 60 bp dsDNA: oligos Cy5-X0m1 and XM3 were annealed by heating to 95 °C, then slowly cooling to 4 °C at a rate of 1 °C/min. 60 bp dsDNA was stored at −20 °C.

To generate 30 bp static Holliday junction used in ATPase assays: oligos Cy5-XOs1, XOs2, XOs3, and XOs4 were annealed by heating to 95 °C, then slowly cooling to 4 °C at a rate of 1 °C/min.

Glycerol (final concentration 5%) was added to the annealed Holliday junctions, which were then gel-purified. The annealed substrates were resolved by 6% PAGE at 100 V for 1 h in 1× TBE. In all, ~100 µl Holliday junctions were excised and eluted overnight by passive diffusion into TMgN buffer A (10 mM Tris pH 8.0, 1 mM MgCl$_2$, 50 mM NaCl). The recovered HJ was then concentrated by precipitation and resuspension in TMgN buffer A. Static HJ was stored at −20 °C.

For 50 bp HJ, pairwise annealing of oligos Xam1-3Dabcyl and Cy3-Xam2, and Xam3 and Xam4, was performed. The two pairs were then incubated together for 1 h at room temperature. Glycerol (final concentration 5%) was added to the solution, before the annealed structures were gel-purified by 6% PAGE. In all, ~100 µl Holliday junctions were excised and eluted overnight by passive diffusion into TMgN buffer B (10 mM Tris pH 8.0, 10 mM MgCl$_2$, 50 mM NaCl). Recovered migratable HJ was stored at 4 °C.

### Enzyme-coupled ATPase assays

Reaction solutions containing 4 nM of FANCM translocase or full-length FANCMc, and 0, 4, 8, 16, 32, 64 and 128 nM of 30 bp non-migratable HJ, or 60 bp dsDNA, were prepared in 1x assay buffer (20 mM Tris pH 7.4, 75 mM NaCl, 5% v/v glycerol, 1 mM DTT, 0.1 mM EDTA, 0.005% NP-40, 1 mM MgCl$_2$) and 1× ATPase cocktail (0.2 mM NADH, 2 mM Phospho(enol)pyruvic acid tri(cyclo-hexylammonium) salt (sigma life science P7252-500MG, made up to 100 mM stock in milliQ water), 6 U/ml Pyruvate kinase (Roche, ref: 10128155001), 9 U/ml L-lactate dehydrogenase (Roche, ref: 10127230001). Solutions were transferred to a 384-well assay plate (Corning, black with clear flat bottom; ref: 3764). Reactions were incubated at 37 °C for 2 h in a BMG CLARIOstar Plus Plate Reader. During this time, absorbance at 340 nm was monitored as a measure of NADH consumption over time. In total, 1 mM ATP and 2 mM MgCl$_2$ was injected ~10 min into the incubation time to initiate the reactions. The reaction rate for the linear phase of each reaction was determined, and a plot against [HJ] was made. Reactions were performed in triplicate, and Michaelis–Menten enzyme kinetics analyses were performed using GraphPad Prism 10.

### Real-time branchpoint translocase assays

Reaction solutions containing 10 nM of FANCM translocase (WT or indicated mutant) or full-length FANCMc (WT or indicated mutant), and 50 nM of 50 bp migratable HJ were prepared in reaction buffer (20 mM Tris pH 7.4, 75 mM NaCl, 5% v/v glycerol, 1 mM DTT, 0.1 mM EDTA, 0.005% NP-40, 1 mM MgCl$_2$). Solutions were transferred to a 384-well polypropylene microplate (Greiner Bio One, ref: 781209), before reactions were incubated at 37 °C for 1.5 h in a BMG LABTECH CLARIOstar Plus Microplate Reader. During this time, fluorescence by Cy3 (excitation 530 nm, emission 580 nm) was monitored as a measure of branch migration over time. At $t = 9$ min, 1 mM ATP was injected into the reactions to initiate ATP hydrolysis and branch migration. Titration experiments were performed as described above, with 0, 1.25, 2.5, 5, 10, 20, or 40 nM of full-length FANCMc. 40 nM of FANCMc was used in reactions where ATP was replaced with either 1 mM ATPγS or 1 mM AMP-PNP. Following the

incubation period, 10 µl samples (of the indicated reactions) were deproteinized for 10 min at 37 °C with 3 mg/ml Proteinase K in branch migration loading dye (final concentrations 50 mM EDTA, 0.3% SDS, 10% glycerol, 0.17 mg/ml bromophenol blue). Reactions were resolved by electrophoresis through 6% polyacrylamide gel in 1× TBE at a constant 100 V for 1 h, then SYBR Gold post-stained for 20 min in 1× TBE. Results were imaged using the Invitrogen iBright CL750 Imager.

Coupled translocase/ATPase assays were performed as per the real-time branchpoint translocase assay, with the addition of 1× assay cocktail (0.2 mM NADH, 2 mM Phospho(enol)pyruvic acid tri(cyclo-hexylammonium) salt (sigma life science P7252-500MG, made up to 100 mM stock in milliQ water), 6 U/ml Pyruvate kinase (Roche, ref: 10128155001), 9 U/ml L-lactate dehydrogenase (Roche, ref: 10127230001) in the assay conditions. In addition, fluorescence by NADH (excitation 380 nm, emission 470 nm) was monitored as a measure of NADH/ATP consumption over time.

### FANCM translocase domain crystallization

Human FANCM$_{(73-644)}$ was in buffer containing 20 mM HEPES pH 7.4, 100 mM NaCl, 1 mM TCEP, and 5 mM MgCl$_2$. For crystallization, the protein was complexed with a Hairpin15 (5′-GGTATGAG-CACTGCTTAGGCAGTGCTCATACCGCATGGAGCTG-3′). Crystals were grown by hanging drop vapor diffusion at room temperature by mixing the protein-DNA complex with crystallization solution (0.2 M sodium chloride, 0.1 M HEPES pH 7.0, 32.5% v/v PEG 400) in a 2:1 ratio. X-ray diffraction data were collected at beamline I03 at Diamond Light Source using an Eiger 2 XE 16 M detector. Data were collected at a wavelength of 0.97628 Å to a resolution of 2.20 Å. The structure was solved by molecular replacement using an AlphaFold-predicted model of FANCM as the search model. Structure refinement was carried out using REFMAC5 (Murshudov et al, 2011).

### FANCM$_{CTD}$:FAAP24 crystallization

HPLC-purified oligonucleotides (SA1: 5′-TAC GCA TCA TCG CTC GGT TTT-3′ and SA2: 5′-TTTT CCG AGC GAT GAT GCG TA-3′) (Jena Bioscience) were dissolved to 1 mg/ml in annealing buffer (5 mM phosphate pH 6.8, 200 mM KCl), mixed 1:1 with complementary strands, heated to 95 °C for 5 min and slow-cooled overnight. Annealed oligonucleotides were purified using mono-Q columns (1 or 5 ml, GE Healthcare) with an ÅKTA FPLC system, eluting with a 0–1 M KCl gradient over 25 column volumes. DNA-containing fractions were verified by agarose gel electrophoresis, pooled, buffer exchanged into water using NAP-25 columns, and concentrated to 250 mM using 5 kDa MWCO vivaspin concentrators. For crystallization, oligonucleotides were mixed with purified FANCM-$_{CTD}$-FAAP24 protein at 3:1 molar ratio, and nucleated as crystals in Nextal PEG/ion screen reagent H7: 25% PEG3350, 0.1 M di-ammonium monohydrogen phosphate. X-ray diffraction data were collected at Diamond Light Source. The diffraction pattern had multiple lattices at low resolution, but we were able to process the diffraction data using DUI2 (Winter et al, 2018). The final model shows a number of adducts associated with the presence of ß-mercaptoethanol used in buffers for protein storage. For example, several cysteines are oxidized to S,S-(2-HYDROXYETHYL)THIOCYSTEINE, known Cys-CEM.

## AlphaFold modeling

Alphafold modeling was conducted using AF3 (www.alpha foldserver.com) uploading the sequences for full-length human FANCM and FAAP24 and splayed arm DNA sequences Data as presented do not show FANCM residues 1–74 or 645–1780 which mostly displayed low pLLD scores due to disorder. These sequences are however present in the modeling files as deposited to www.modelarchive.org accession number ma-j6afp.

For AlphaFold modeling shown in Fig. 7, full-length FANCM, or FANCM 900–1100 was uploaded to AF3, together with sequences for full-length FANCC, FANCF and FANCE residues 1–181 (lacking the FANCD2-binding C-terminal domain). These modeling files are deposited to www.modelarchive.org accession number ma-ao706.

## Cell-based experiments

HEK293 and HCT116 and derivative lines were maintained as adherent monolayers in DMEM (Sigma Aldrich) containing 10% heat-inactivated fetal bovine serum (Bovogen), at 37 °C in a humidified atmosphere of 5% carbon dioxide.

HEK293 complementation experiments: We previously established stable FANCM-depleted cell lines using a targeted shRNA$^{mir}$ sequence (V2HS_203688, targeting the sequence 5′-CGTTCTACTCCAA-GAGTTAA-3′, Open Biosystems) in the 293-FLPin system, which allows doxycycline-inducible expression of siRNA-resistant FANCM variants (Collis et al, 2008; Deans and West, 2009). The mutations K117R, I1827D, I1827D/V1847D, R1931X were introduced into pDEST-FRT-3xFlag-FANCM by Phusion site-directed mutagenesis (Thermo Fischer). These sequences were then stably integrated into the HEK293-FLPin-FANCM$^{shRNAmir}$ cells using cotransfections with pOG44 (Thermo Fisher) and selection in 500 µg/ml Hygromycin B (Thermo Fisher). HEK293-FLPin-FANCM$^{shRNAmir}$ with empty vector inserted at the FLPin site, or HEK293-FLPin cells with a non-targeting shRNA$^{mir}$ sequence were used as a control in all experiments.

HCT116 FANCM−/− complementation experiments: HCT116 FANCM−/− cells were a kind gift of Li Lei, and have been previously characterized (Wang et al, 2013). For complementation studies, we utilized baculoviruses containing a dual-specificity promoter enabling expression in both insect and mammalian cells(Mansouri et al, 2016). Viruses were identical to those used for protein purification (described above). Cells were seeded at $5 \times 10^5$ cells per well in 6-well plates and transduced with baculovirus at a multiplicity of infection (MOI) of 50 in the presence of 1× BacMam enhancer (Invitrogen), followed by centrifugation at $600 \times g$ for 30 min. After 6 h of incubation, the virus-containing medium was replaced with fresh medium, and cells were allowed to recover for 12 h before proceeding with immunofluorescence and drug sensitivity assays. Control samples were treated with either empty baculovirus or insect expression medium alone. Equivalent expression levels of protein variants were confirmed by western blot analysis.

For immunoprecipitation (IP), cells were lysed in buffer K100 (50 mM KPO$_4$ [pH 7.5], 100 mM NaCl, 10% glycerol, 1 mM DTT, 1 mM MgCl$_2$, 0.1% Triton X-100, 5 mM NaF, complete protease inhibitor) containing 50 U/ml benzonase (Promega) for 2 h at 4 °C. Following lysis, the NaCl concentration was increased to 250 mM, and lysates were cleared by centrifugation at $16,000 \times g$ for 30 min. The supernatant was immunoprecipitated using 20 µl α-Flag M2

agarose (Sigma). After 3 h of mixing, beads were washed four times with K250 buffer and once with 50 mM NH4(CO$_3$)$_2$. Proteins were eluted with 500 mM NH4OH (pH 11.0), 0.5 mM EDTA, dried in a speedvac, and resuspended in 1× LDS loading buffer (Invitrogen) followed by Western blotting for the indicated proteins. Antibodies used were anti-FANCE (Bethyl A302-125A), anti-FANCA (A301-980A), anti-FANCM (CV5.1 ref (Vuono et al, 2016)) and anti-FAAP24 (SWE94 (ref Ciccia et al, 2007)).

Immunofluorescence microscopy was performed using cells grown on eight-well chamber slides (Millicell EZ slide, Sigma). For HEK293 cells, slides were pre-coated with 2.5 µg/ml polyethyleneimine. Cells were seeded at $3 \times 10^4$ cells per well and grown overnight before treatment with 40 ng/ml mitomycin C for 7 h. After PBS washing, cells were fixed with 4% paraformaldehyde for 20 min, followed by cold methanol for 1 min. Cells were permeabilized with 0.3% Triton X-100 in PBS, blocked with 10% FBS and 0.1% NP-40 in PBS, and incubated overnight at 4 °C with primary antibodies against FANCD2 (Novus NB100-182, 1:1000) and γH2AX (Upstate 16-193, 1:1000). After washing in PBS containing 2% FBS, cells were incubated with Alexa Fluor-conjugated secondary antibodies (1:1000) for 2 h at room temperature. Slides were mounted using Dako Fluorescence Mounting Medium containing DAPI (0.1 µg/ml). Images were collected using a Zeiss confocal microscope at ×63 optical magnification and analyzed using CellProfiler software.

## Data availability

Requests for further information and resources should be directed to and will be fulfilled by the corresponding, Andrew Deans (adeans@svi.edu.au). All unique/stable reagents generated in this study are available from the corresponding author with a completed materials transfer agreement. Structures presented in this paper have been deposited at the Protein Databank (FANCM translocase: https://www.rcsb.org/structure/9EL5, FANCM-CTD:FAAP24:splay DNA, https://www.rcsb.org/structure/9HJO) or the Model archive (https://modelarchive.org with accession ma-j6afp (FANCM:-FAAP24) and ma-ao706 (FANCM:FANCC:FANCE:FANCF)).

The source data of this paper are collected in the following database record: biostudies:S-SCDT-10_1038-S44318-025-00468-3.

## Peer review information

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

## Acknowledgements

This research was performed on Wurundjeri land, funded by an Australian Government National Health and Medical Research Council grant (GNT1181110 to AJD and RBD) and core funding from Cancer Research UK (via the Francis Crick Institute CC2068 to NQM). RBD was supported in part by a fellowship from the National Breast Cancer Foundation Australia. We acknowledge the support of the Victorian Government's OIS Program. Thanks to Brenda Schulman for advice on ATP analogs and ubiquitination, Steve West and Weidong Wang for reagents and Jörg Heierhorst and Wayne Crismani for valuable discussions.

## Author contributions

**Rohan Bythell-Douglas**: Conceptualization; Formal analysis; Supervision; Funding acquisition; Investigation; Methodology; Writing—original draft; Project administration; Writing—review and editing. **Sylvie van Twest**: Conceptualization; Formal analysis; Investigation; Visualization; Methodology; Writing—original draft; Writing—review and editing. **Lara Abbouche**: Conceptualization; Formal analysis; Investigation; Visualization; Methodology; Writing—original draft; Writing—review and editing. **Elyse Dunn**: Formal analysis; Investigation; Methodology; Writing—review and editing. **Rachel J Coulthard**: Formal analysis; Visualization; Methodology; Writing—review and editing. **David C Briggs**: Validation; Visualization; Writing—review and editing. **Vincent Murphy**: Validation; Investigation; Methodology; Writing—review and editing. **Xinxin Zhang**: Investigation; Writing—review and editing. **Winnie Tan**: Investigation; Methodology; Writing—review and editing. **Sarah Henrikus**: Resources; Investigation; Writing—review and editing. **Dongming Qian**: Resources; Supervision; Investigation. **Yin Wu**: Conceptualization; Resources; Investigation; Writing—review and editing. **Jana Wolf**: Conceptualization;

Investigation; Writing—review and editing. **Laurent Rigoreau**: Conceptualization; Supervision; Investigation; Writing—review and editing. **Shabih Shakeel**: Conceptualization; Formal analysis; Supervision; Funding acquisition; Investigation; Methodology; Writing—review and editing. **Kathryn L Chapman**: Conceptualization; Supervision; Methodology; Project administration; Writing—review and editing. **Neil Q McDonald**: Conceptualization; Formal analysis; Supervision; Funding acquisition; Visualization; Writing—original draft; Project administration; Writing—review and editing. **Andrew J Deans**: Conceptualization; Data curation; Formal analysis; Supervision; Funding acquisition; Investigation; Visualization; Methodology; Writing—original draft; Project administration; Writing—review and editing.

Source data underlying figure panels in this paper may have individual authorship assigned. Where available, figure panel/source data authorship is listed in the following database record: biostudies:S-SCDT-10_1038-S44318-025-00468-3.

## Funding

## Disclosure and competing interests statement

JW, LR, and KC are paid employees of Tessellate Bio. AJD is a member of the scientific advisory board of Tessellate Bio. Conflicts are managed by St Vincent's Institute. The remaining authors declare no competing interests.

# Expanded View Figures

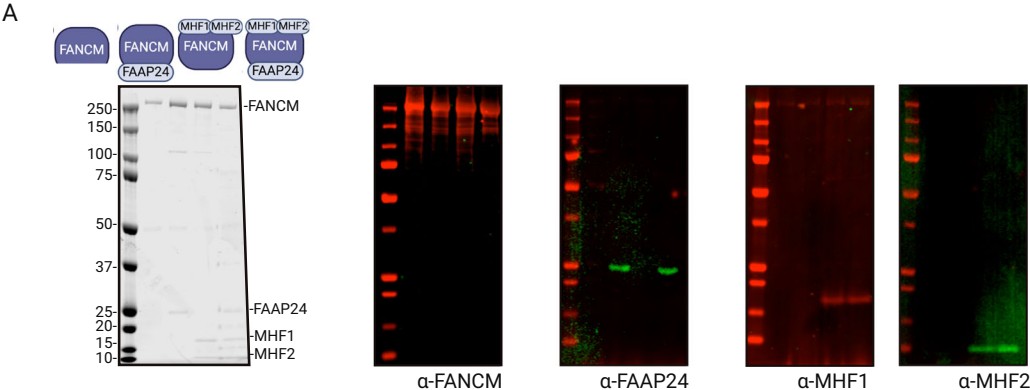

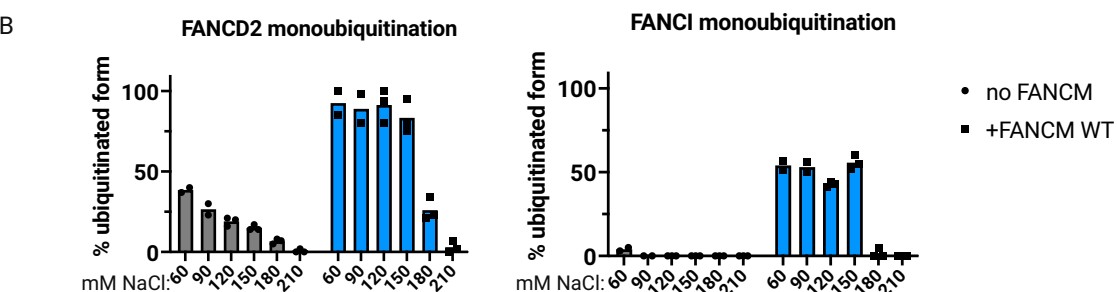

**Figure EV1. Purified FANCM complexes overcome the salt inhibition of FANCD2 and FANCI monoubiquitination by FA core complex.**

(A) Purified FANCM complexes, shown by Coomassie blue staining (as per Fig. 1B). Western blotting reveals that each subunit is present. (B) FANCD2:FANCI monoubiquitination reactions conducted in increasing concentrations (60, 90, 120, 150, 180 and 210 nM) NaCl without FANCM (gray bars) or with 100 nM FANCM (blue bars). Reactions were stopped at 30 min and ubiquitinated form was quantified. Results from 2 (60 and 90 nM) or 3 (other concentrations) independent experiments, individual values, mean ± SD shown.

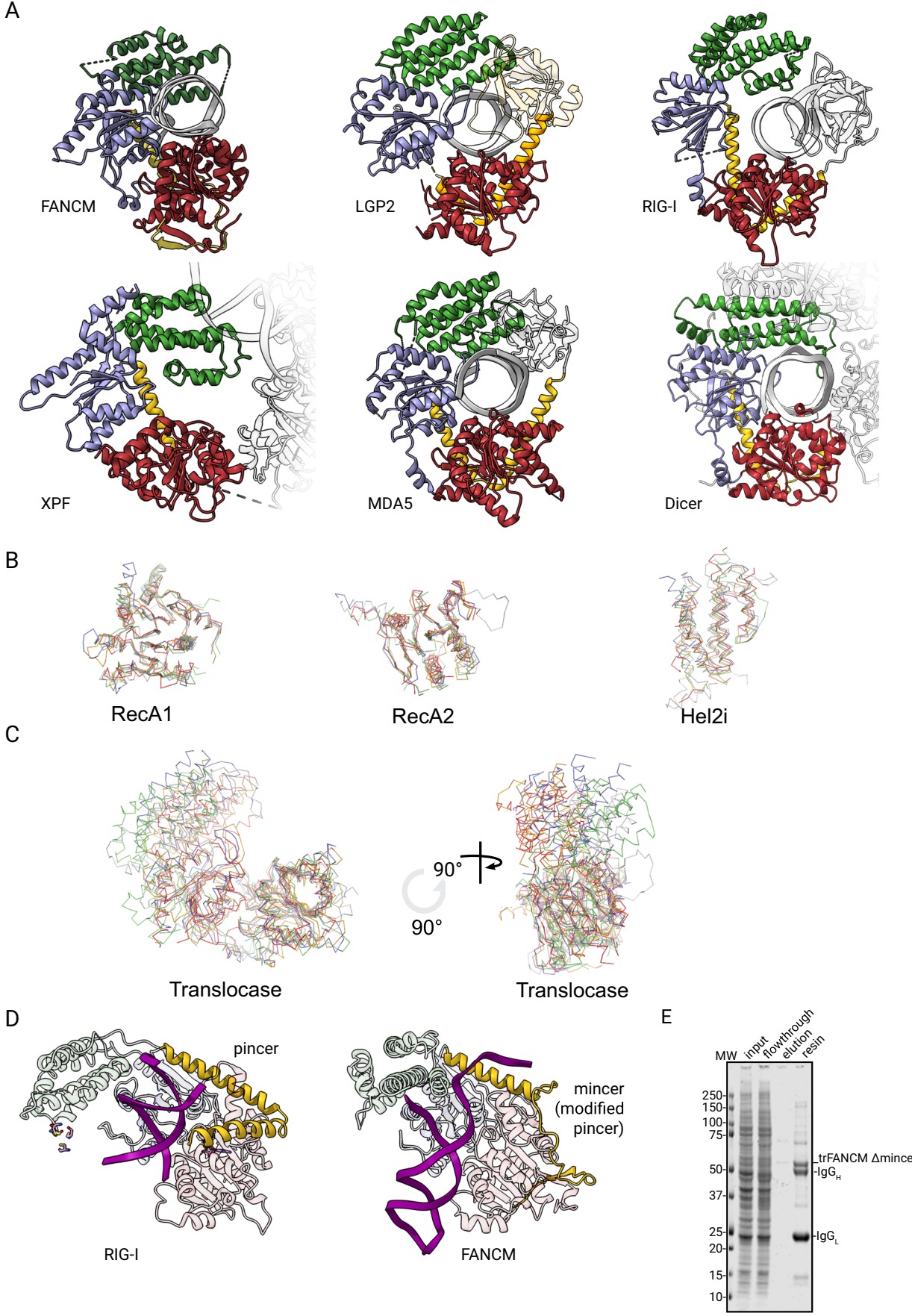

**Figure EV2.   Similarities and differences between FANCM translocase domain and structurally related proteins.**

(A) Related translocase domains with structures in the PDB: LGP2 1–482 (Uchikawa et al, 2016), RIG-I 236–744 (Luo et al, 2011), XPF 1–631 (Jones et al, 2020), MDA-5 308–836 (Yu et al, 2018) and Dicer 8-508 (Deng et al, 2023). All related structures are bound to dsRNA and no ATP analog (except for XPF which is bound to dsDNA). (B) Domains were superposed using "SUPERPOSE" from within the CCP4 program suite, using secondary structure matching for the specified residue ranges. Structures are depicted in ribbon representation in pymol software. FANCM = red, LGP2 = orange, RIG-I = green, MDA-5 = blue, Dicer = gray. (C) full translocase domains overlayed (XPF excluded) superposed using the RecA2 domain, coloring as in (B, D) specific orientation of RIG-I and FANCM highlighting the "mincer" domain of FANCM compared to the "pincer" domain of RIG-I. (E) trFANCM-Δmincer (FANCM 80–590) is mostly insoluble and remains stuck on resin during Flag affinity purification.

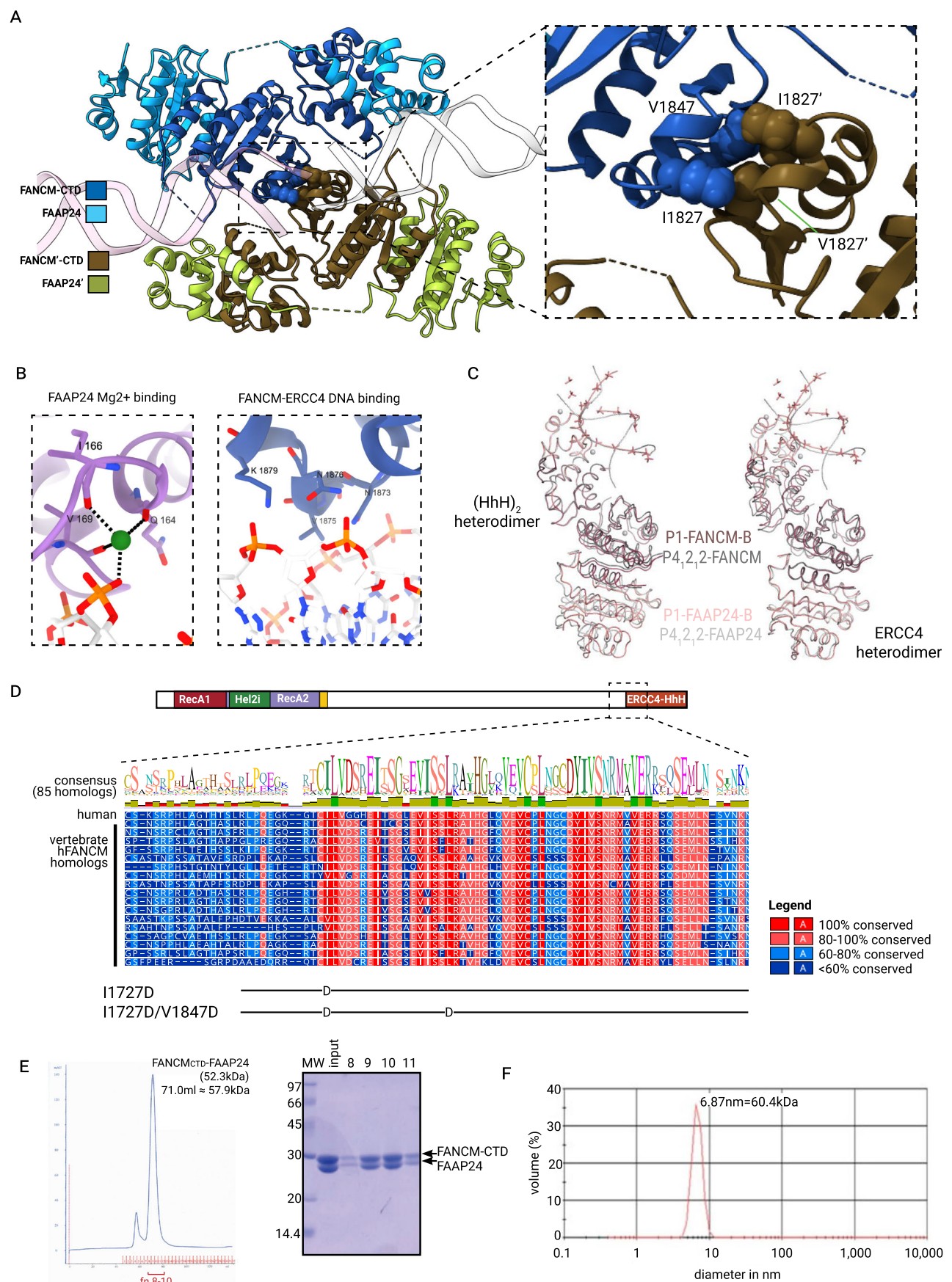

**Figure EV3.   Detailed view of FANCM-CTD:FAAP24 structure and conservation.**

(A) Dimeric arrangement of FANCM-CTD:FAAP24 complexes in P4,2,2 crystal. The interface involves helix 1 and 2 and beta-strand 2 of the two different molecules, which form an alternative unit cell, and residues at the interface are highly conserved within FANCM. Key residues include I1827, which is highly conserved in FANCM, packing against V1847 and P1849. The interface has 2-fold non-crystallographic symmetry, and in general residues at this interface have a highly similar positions and the same rotamer on each side of the interface. The one exception is R1838, which is only modeled in copy B. (B) zoom in of left:FAAP24 (Mauve) and $Mg2+$ (green) binding site, and right: additional FANCM-DNA-binding site. Key residues shown by numbering. Phosphate backbone of DNA in orange/red. Refer to main text for details. (C) Superposition using the (HhH)2 and ERCC4 heterodimers separately shows that the orientation of the two domains is slightly different in the two crystal forms, suggesting that the ERCC4-$(HhH)_2$ interface may be subtly different. This change in orientation is, however, very slight and the DynDom server (Hayward and Berendsen 1998), which allows easy calculation of domain movements in different crystal forms was unable to detect it. The core hydrophobic residues involved in this intramolecular FANCM interface are identical, consistent with it being a conserved feature present in solution. (D) Conservation of the region of FANCM ERCC4 domain N-terminal boundary across 85 vertebrate homologs using the consurf server (Ashkenazy et al, 2016). (E) Size exclusion chromatography reveals a peak corresponding to size of heterodimer. (F) Dynamic light scattering (DLS) shows size expected for a heterodimer in solution.

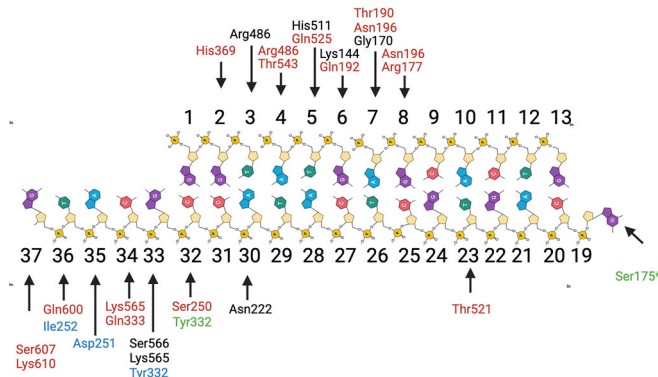

**Figure EV4.  FANCM translocase contacts with DNA.**

DNA bases are numbered from 5′ to 3′ end of co-crystallized oligonucleotide. Amino acids of FANCM contacting DNA are shown in black (backbone-backbone contacts), red (sidechain-backbone contacts), blue (backbone-to-base contacts) and green (sidechain-base contacts).

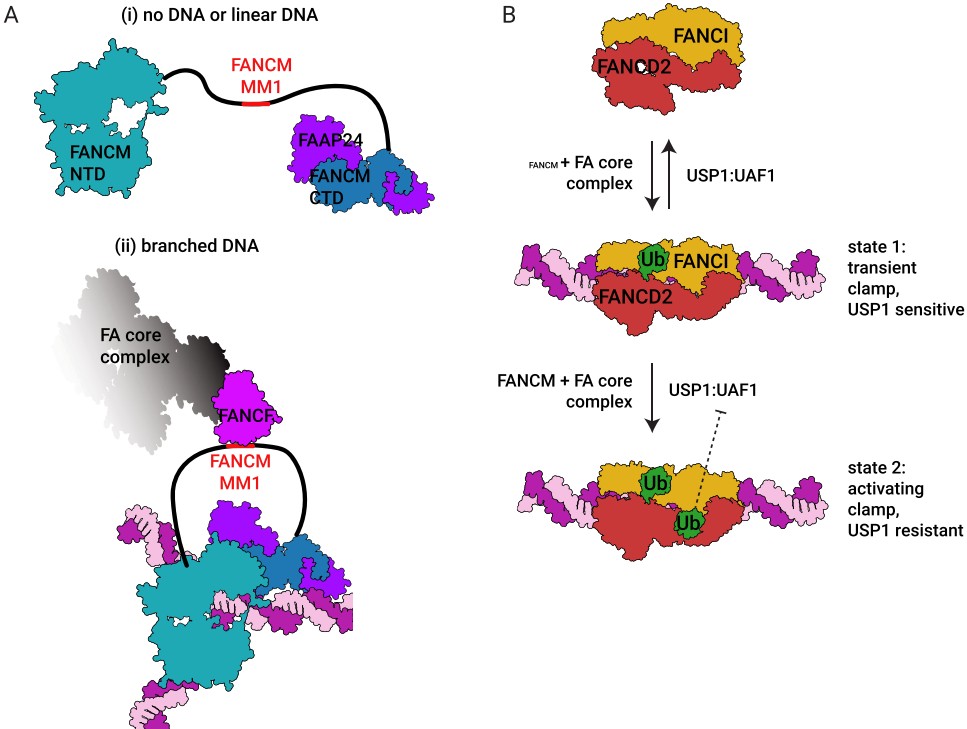

**Figure EV5. Models based on data generated in this manuscript aligned to existing literature.**

(A) FANCM also exists in at least 2 states: (i) a free state where the N-terminus and C-terminus are not associated, and the protein is not activating the ubiquitination dependent pathway and (ii) a DNA bound state where the N- and C-terminus come together which activates the complex to monoubiquitinate FANCD2, but drive the enzyme to sequentially and rapidly ubiquitinate FANCI. In this way, FANCM drives the quick progression of the transient ID2 clamp shown in (A) to the activating clamp required for DNA repair signaling. The motor of FANCM also becomes activated, to stabilize the fork structure, required for the downstream repair steps. (B) Dual monoubiquitination converts FANCD2:FANCI from a transient to stable clamp on DNA.

