## [Peer Review File · The EMBO Journal]

Structural basis of Fanconi Anemia pathway activation by FANCM

Rohan Bythell-Douglas, Sylive van Twest, Lara Abbouche, Elyse Dunn, Rachel Coulthard, David Briggs, Vincent Murphy, Xinxin Zhang, Winnie Tan, Sarah Henrikus, Dongming Qian, Yin Wu, Jana Wolf, Laurent Rigoreau, Shabih Shakeel, Kathryn Chapman, Neil McDonald, and Andrew Deans

Corresponding author(s): Andrew Deans (adeans@svi.edu.au)

Review Timeline:

Submission Date:	11th Mar 25
Editorial Decision:	1st Apr 25
Revision Received:	23rd Apr 25
Editorial Decision:	12th May 25
Revision Received:	14th May 25
Accepted:	14th May 25

Editor: Hartmut Vodermaier

Transaction Report:

Prof. Andrew J Deans
St Vincent's Institute
Genome Stability Unit
9 Princes St
Fitzroy, VIC 3065
Australia

1st Apr 2025

Re: EMBOJ-2025-120753-T
Structure and mechanism of FANCM reveals separable functions in genome maintenance

Dear Andrew,

Thank you again for submitting your structural study on separable genome maintenance functions of FANCM to The EMBO Journal. I have now received reports from three expert referees, copied below for your information. As you will see, all reviewers appreciate the general interest and overall quality of the work, but nevertheless raise several issues that would need to be addressed prior to publication. Pending adequate revisions in response to these comments, we would be happy to consider the study further for The EMBO Journal.

Please be reminded that our single-major-revision-round policy makes it important to diligently respond to each referee point at the time of resubmission; therefore, although the majority of the referees' concerns are presentational, please do not hesitate to contact me with any questions you may have regarding any additional analyses suggested. We would also be open to extension of the regular three-months revision period if needed; our 'scooping protection' (meaning that competing work appearing elsewhere in the meantime will not affect our considerations of your study) would of course remain valid also throughout such an extension.

Further information on preparing, formatting and uploading a revised manuscript can be found below and in our Guide to Authors. Thank you again for the opportunity to consider this work for The EMBO Journal, and I look forward to receiving your revised manuscript in due time.

With kind regards,

Hartmut

9) To facilitate reproducibility and cross-laboratory adoption of methodologies, please structure the Materials & Methods section as outlined in our guide to authors, including a completed Reagents and Tools Table that can be downloaded from our author guidelines as well (<https://www.embopress.org/page/journal/14602075/authorguide#structuredmethods>).

10) Digital image enhancement is acceptable practice, as long as it accurately represents the original data and conforms to community standards. If a figure has been subjected to significant electronic manipulation, this must be clearly noted in the figure legend and/or the 'Materials and Methods' section. The editors reserve the right to request original versions of figures and the original images that were used to assemble the figure. Finally, we generally encourage uploading of numerical as well as gel/blot image source data; for details see: embopress.org/page/journal/14602075/authorguide#sourcedata

At EMBO Press, we ask authors to provide source data for the main manuscript figures. Our source data coordinator will contact you to discuss which figure panels we would need source data for and will also provide you with helpful tips on how to upload and organize the files.

In the interest of ensuring the conceptual advance provided by the work, we recommend submitting a revision within 3 months (30th Jun 2025). Please discuss the revision progress ahead of this time with the editor if you require more time to complete the revisions. Use the link below to submit your revision:

Link Not Available

Referee #1:

Main finding, through structural and biochemical investigation is that two domains of FANCM, one at the N-terminus, and one more C-terminal have different activities. They show that adding FANCM to FANCI/FANCD2 monoubiquitination experiments stimulates the amount of ubiquitination, in a manner dependent on DNA structure. They show both domains (ctd including Faap24) bound to DNA, and validate the interactions, and show how these residues coordinate ATPase and branch migration activities. Finally they use patient mutations to demonstrate the role FANCM has in recruiting members of the FA core complex to support the monoubiquitination.

These are important findings that add to the understanding of the FA DNA repair pathway.

Will be interesting to see one day how the domains are organised in the context of the full-length protein, and whether they are entirely independent modules, or whether DNA binding and the context of the core complex regulates the activities.

Data are high quality, not overinterpreted, well-controlled and well presented. The structural data in particular are high quality. Not sure what purpose the pictures of crystals serve, though they are very nice! On second thoughts, they make clear to the reader that the structures are experimental data, rather than alpha fold models, and therefore I recommend leaving them in.

Minor points

Title is misleading - this is not the structure of FANCM, it is the structure of two domains of FANCM

References are missing some of the other contemporaneous papers describing the formation of the clamp - Wang 2020, Nature, Rennie 2020, EMBO Rep

Citations for USP1-UAF1 deubiquitination of FANCI - could also include Lemonidis et al., EMBO Journal, 2023 in the context of USP1's activity towards D2-ub in the doubly-modified ID2 heterodimer.

All Alpha fold models should include pae and plddt plots in the extended data (included for some but not others). I would also recommend uploading the models (particularly of the complex) to a database such as modelarchive, to allow other researchers to scrutinise the chosen models.

PDB files should have the citations (in figure legends) 7KZP (Wang et al., 2021, NSMB); 7ZKQ appears to be an electron transport protein, I suspect the authors mean 7KZQ, also Wang et al., 2021, NSMB.

Referee #2:

The manuscript from the Deans laboratory solves some of the outstanding questions in how FANCM participates in activation of the Fanconi anemia pathway during DNA interstrand crosslink repair. They show that the interaction with the MHF1 and 2 proteins are not necessary for stimulation of ubiquitination which is prevalent in models of the pathway activation. They model the known C terminal portion of FANCM interacting with FANCF (PMID 20064461) and show that this interaction is necessary for stimulation of the FANCI-D2 ubiquitination by the core complex. The authors show convincing biochemical experiments to support these and other data and do a good job with testing their structural findings with appropriate mutants. I believe that this will be an important work for the Fanconi anemia field and understanding breast cancer. Below are some suggestions for improvements:

1. It would be important to discuss the FANCM activity of traversing the ICL (described by the Seidman lab (PMID 24207054) and the authors findings about different parts of FANCM that might be important for such activity. Ideally, they would test different mutants in that assay, although I realize this might not be within the scope of this paper.
2. Can the authors comment if breast tumors have FANCM LOH that would leave only the mutant FANCM protein being expressed
3. The authors discuss that the infertility of patients with FANCM mutations is different than infertility of FA patients. I do not believe they have enough data to support that. I believe that detailed genotype-phenotype studies in patients would be necessary to make that conclusion.
4. The quantification in Fig4C is misleading. The vector control and the two mutants still appear to have foci (as they should, since FANCM deficiency gives a partial inhibition of ID2 Ub) but they are much less striking suggesting fewer UB ID2 complex at the damage sites. The authors could say that they scored bright foci.
5. In general, the D2 foci will be very difficult to see in a printed form. I suggest using grey scale for those. It also appears that some mutants may have effects on levels of D2 (Fig6F +K117R). Is that the case on a western blot?
6. I would add the findings about MHF1 and 2 into the abstract.

Minor:

1. Where is the poly dIdC data shown? It is mentioned on page 2, line 80-82.
2. The gamma H2AX staining is very high in 4C suggesting a lot of damage, high MMC levels were used for this experiment and this is why even WT FANCM cells showing so much gammaH2AX.
3. I find some of the figure legends lacking in detail. For example, were Fig1D experiments done in the presence of DNA?
4. Figure legend for Fig 6 is completely wrong- it is copied from Figure 5 legend.
5. Figure 6B the filled and open circles are overlapping. I could figure out what is what, but the authors might want to have four different colors/symbols. Figure 6H and 7J- the lines are very thick- different in any other graph in those figures. they might want to change the thickness.
6. Page 5 right column, line 109, "it" in " it is absolutely required..." means FANCF? Needs clarification

Referee #3:

FANCM has multiple functions in the maintenance of genome integrity. It functions as a complex with FAAP24, MFH1 and MFH2. FANCM contributes to the mono-ubiquitination of FANCI/FANCD2 (a core step in the activation of the FA pathway), and it is simultaneously a motor protein that acts during various steps of DNA repair and replication stress response.

In the beginning of the paper, it is shown in reconstituted assays that FANCM promotes FANCI/FANCD2 mono-ubiquitination (as known from before) and that FAAP24 is not required for this process. The story then follows by describing the structures of the N and C-terminal domains of FANCM, with DNA, and structural modeling that identified an interface between the N- and C-terminal domains. This part of the paper was the weakest, in my opinion, some data were overinterpreted and the advance that these data bring to the field, with respect to published literature, was not always clear. The story then follows strongly by showing that some identified DNA contact points, as well as the ATPase activity of FANCM are not necessary for

FANCI/FANCD2, but are instead needed e.g. for cellular response to DNA damage. Hence, the key functions of FANCM are separable. Overall, the manuscript contains a lot of high-quality data, but the story can be improved by improving flow and clarifying the conceptual advance.

Specific comments:

Fig. 1E: It is noted that FANCM promotes monoubiquitination to the point that it becomes resistant to the USP1-UAF1 deubiquitinase. While I see what the authors conclude concerning the modified upper band, there is a gradual decrease of the unmodified band (both with DANCD2 and FANCI) with increasing time of incubation in the presence of USP1-UAF1 (lanes 8-10). What is happening to the protein? Also, the legend notes that statistics is shown, but the size of the graphs makes any error bars invisible. Y axis is unlabeled.

Fig. 1G: Same comment on quantification as above applies to panel G. The labeling makes the experiment somewhat difficult to understand. Legend notes "complete reaction", while it is not immediately clear what is meant; the legend should note what is quantified; also, the text refers to lanes 12-14, but no lane numbers are given. Do the authors know that their preparation of the complex containing MHF1 and MHF2 increases motor activity of FANCM? The conclusion on the separation of function can only be made with this control.

Figure 2-3: Please include residue numbers on the primary structure cartoons. Figure 3 shows the structure of the C-terminal region of FANCM together with FAAP24. A similar complex has already been structurally analyzed in Ref32: "Structural insights into the functions of the FANCM-FAAP24 complex in DNA repair". While Ref32 is cited, it was in the general context that FAAP24 is needed for FANCM function. The advance of the new structure should be made more clear with respect to the previous data.

Figure 4: While Figure 2 and Figure 3 show crystal structures of FANCM's N-terminal and C-terminal domains with DNA in isolation, Figure 4 combines these data with AF3 modeling to identify potential interface sites that bring the N- and C-terminal domains together. A mutation is analyzed, which is predicted to disrupt the interface (i.e. ternary structure), and which results in defects in cell survival and foci experiments. While the model is most likely correct, I do not see how these data support the conclusion: "These findings reveal FANCM's sophisticated DNA damage recognition system, where the motor domain identifies branch points while the C-terminal domains provide additional grip and positioning." Along the same lines, it is noted "The discovery that FANCM's domains physically interact suggested they might functionally cooperate in migrating these branched DNA structures." It is known from the previous literature (e.g. Ref32) that the C-terminal domain targets FANCM to chromatin, and as the motor domain is in the N-terminus, it is not clear how the data conceptually advance the field.

Could the authors demonstrate whether the discovered separation of function mutations differently impact the cellular functions of FANCM? The discovery that the ATPase and DNA binding mutants are not needed for FANCI/FANCD2 mono-ubi but are required for branch migration is exciting, but I feel that the story somehow falls short to demonstrate how these two (newly discovered) distinct processes impact on the processes in which FANCM promotes genome stability. While this may be above the scope of a revision, I feel that this would really bring the field forward.

We thank the reviewers for their overall positive and constructive feedback on our manuscript. In response to the submission criteria for *EMBO Journal*, we have revised the abstract, introduction and discussion sections to align with the journal's style requirements. Below, we also provide a point-by-point response to each referee's specific comments:

Referee #1:

Main finding, through structural and biochemical investigation is that two domains of FANCM, one at the N-terminus, and one more C-terminal have different activities. They show that adding FANCM to FANCI/FANCD2 monoubiquitination experiments stimulates the amount of ubiquitination, in a manner dependent on DNA structure. They show both domains (ctd including Faap24) bound to DNA, and validate the interactions, and show how these residues coordinate ATPase and branch migration activities. Finally they use patient mutations to demonstrate the role FANCM has in recruiting members of the FA core complex to support the monoubiquitination. These are important findings that add to the understanding of the FA DNA repair pathway.

Will be interesting to see one day how the domains are organised in the context of the full-length protein, and whether they are entirely independent modules, or whether DNA binding and the context of the core complex regulates the activities. Data are high quality, not overinterpreted, well-controlled and well presented. The structural data in particular are high quality. Not sure what purpose the pictures of crystals serve, though they are very nice! On second thoughts, they make clear to the reader that the structures are experimental data, rather than alpha fold models, and therefore I recommend leaving them in.

We thank the reviewer for this appreciative summary.

Minor points

Title is misleading - this is not the structure of FANCM, it is the structure of two domains of FANCM

*We have changed the title to reflect this. It is now "**Structural basis of Fanconi Anemia pathway activation by FANCM**".*

References are missing some of the other contemporaneous papers describing the formation of the clamp - Wang 2020, Nature, Rennie 2020, EMBO Rep
Citations for USP1-UAF1 deubiquitination of FANCI - could also include Lemonidis et

al., EMBO Journal, 2023 in the context of USP1's activity towards D2-ub in the doubly-modified ID2 heterodimer.

We have added in these references for completeness.

All Alpha fold models should include pae and plddt plots in the extended data (included for some but not others). I would also recommend uploading the models (particularly of the complex) to a database such as modelarchive, to allow other researchers to scrutinise the chosen models.

All models of the paper have been deposited to Model Archive. This was indicated in the materials and methods section of the previous version, but we have now added additional reference to these deposits in the “resource availability” section.

PDB files should have the citations (in figure legends) 7KZP (Wang et al., 2021, NSMB); 7ZKQ appears to be an electron transport protein, I suspect the authors mean 7KZQ, also Wang et al., 2021, NSMB.

We have now added all citations to PDB files cited in figure legends.

Referee #2:

The manuscript from the Deans laboratory solves some of the outstanding questions in how FANCM participates in activation of the Fanconi anemia pathway during DNA interstrand crosslink repair. They show that the interaction with the MHF1 and 2 proteins are not necessary for stimulation of ubiquitination which is prevalent in models of the pathway activation. They model the known C terminal portion of FANCM interacting with FANCF (PMID 20064461) and show that this interaction is necessary for stimulation of the FANCI-D2 ubiquitination by the core complex. The authors show convincing biochemical experiments to support these and other data and do a good job with testing their structural findings with appropriate mutants. I believe that this will be an important work for the Fanconi anemia field and understanding breast cancer. Below are some suggestions for improvements:

1. It would be important to discuss the FANCM activity of traversing the ICL (described by the Seidman lab (PMID 24207054) and the authors findings about different parts of FANCM that might be important for such activity. Ideally, they would test different mutants in that assay, although I realize this might not be within the scope of this paper.

We have now discussed the possibility of traverse in the introduction, and in a short section of paragraph 2 of the discussion.

2. Can the authors comment if breast tumors have FANCM LOH that would leave only the mutant FANCM protein being expressed.

The study which identified the FANCM MM1 domain mutations did not investigate whether LOH occurred. Not all DNA repair protein mutants require LOH to occur for their to be a genetic effect (eg see ref [10.1016/j.celrep.2018.03.076](https://doi.org/10.1016/j.celrep.2018.03.076)). We are currently working with the Southey/Campbell teams to access cancer material that might be available from these carriers to answer this interesting question.

3. The authors discuss that the infertility of patients with FANCM mutations is different than infertility of FA patients. I do not believe they have enough data to support that. I believe that detailed genotype-phenotype studies in patients would be necessary to make that conclusion.

I think there is a misunderstanding. We haven't proposed that FANCM mutations cause a different type of infertility vs FA patients. In the discussion we only propose that because ATPase defective mutants have been associated with male infertility (but also, so have deletion mutations) but MM1 mutants have been associated with breast cancer there may be genotype-phenotype correlation in FANCM mutant individuals that requires further exploration. However, in a C57Bl6 mouse context, FANCM-deficiency causes only 50% loss of sperm production (see [10.1016/j.xgen.2023.100349](https://doi.org/10.1016/j.xgen.2023.100349)) whereas loss of FA core complex components causes total infertility (eg [10.1002/path.2992](https://doi.org/10.1002/path.2992), [10.1093/hmg/11.24.3047](https://doi.org/10.1093/hmg/11.24.3047), [10.1093/hmg/ddg219](https://doi.org/10.1093/hmg/ddg219) and other. So, it may be worth a future exploration of why this difference exists, and whether the difference in retained fertility is as striking in humans.

4. The quantification in Fig4C is misleading. The vector control and the two mutants still appear to have foci (as they should, since FANCM deficiency gives a partial inhibition of ID2 Ub) but they are much less striking suggesting fewer UB ID2 complex at the damage sites. The authors could say that they scored bright foci.

We have changed the figure legend and text to state that only bright foci were scored.

5. In general, the D2 foci will be very difficult to see in a printed form. I suggest using grey scale for those. It also appears that some mutants may have effects on levels of D2 (Fig6F +K117R). Is that the case on a western blot?

We will discuss with the publishers about whether it is better to change these images to greyscale. The apparent increased brightness of foci in the K117R example image is not reflected in the overall quantification of foci (scored using software) and is due to some minor variability in overall staining intensity seen across the slides.

6. I would add the findings about MHF1 and 2 into the abstract.

We agree that the MHF findings are interesting, particularly because MHF deficiency leads to loss of FANCD2 in cells (eg in [10.1016/j.molcel.2010.01.039](https://doi.org/10.1016/j.molcel.2010.01.039) and [10.1016/j.molcel.2010.01.036](https://doi.org/10.1016/j.molcel.2010.01.036) however we think this is because in MHF1 and 2 deficient cells, FANCM becomes abnormally degraded. Our biochemical data supports this conclusion. However we have a lot to convey in the abstract and only 175 words limit! Instead, we have included some new discussion of the MHF finding in the discussion section, and added MHF as a key word to the manuscript.

Minor:

1. Where is the poly dIdC data shown? It is mentioned on page 2, line 80-82.

Poly dIdC is used as a highly branched DNA molecule, for all of the ubiquitination experiments of the manuscript, except for those in Figure 1C. We have changed the methods section to indicate this.

2. The gamma H2AX staining is very high in 4C suggesting a lot of damage, high MMC levels were used for this experiment and this is why even WT FANCM cells showing so much gammaH2AX.

The amount of MMC used in HEK293 cells with the Fig 4C experiment is equivalent to that previously used to show the role of FANCM in FANCD2 foci formation and is approximately equivalent to the IC50 of the drug in this cell line (see eg [10.1093/hmg/ddp297](https://doi.org/10.1093/hmg/ddp297), [10.1016/j.molcel.2009.12.006](https://doi.org/10.1016/j.molcel.2009.12.006)). The amount used for the experiments of Figure 6 and 7 is lower as the IC50 in HCT116 is closer to 0.1ng/ml..

3. I find some of the figure legends lacking in detail. For example, were Fig1D experiments done in the presence of DNA?

All ubiquitination experiments were conducted in presence of DNA, because ubiquitination is very low on non-DNA bound forms of FANCI:FANCD2 complex. As per the updated materials and methods we have now made it clear that poly-dIdC was used in all experiments.

4. Figure legend for Fig 6 is completely wrong- it is copied from Figure 5 legend.

The figure legends appeared in 2 different places in the original submission (and the end of the text document and under each figure picture). During the copying step to ensure that they matched, we accidentally duplicated the legend. The correct legend is included in the final submission at the end of the main document (and not with the figure, as per EMBO J final submission guidelines). We sincerely apologise for this error.

5. Figure 6B the filled and open circles are overlapping. I could figure out what is what , but the authors might want to have four different colors/symbols.

Upon careful consideration, we've maintained the current presentation format as it deliberately serves our intended scientific communication. The consistent colour scheme (same colour for identical ATP/ATPyS conditions) and fill style (same fill for with/without FANCM) were specifically chosen to emphasise the key finding: the near-perfect overlap between certain experimental conditions. This approach allows readers to immediately grasp that data points with the same fill characteristics cluster together, while data points with the same colour characteristics also cluster together. Using four entirely different colours/symbols would make it more difficult to see this pattern of overlaps, which is central to our interpretation of the results. In the figure legend we made this more clear. We hope this approach maintains scientific clarity while addressing your valid concern about distinguishing between the experimental conditions.

Figure 6H and 7J- the lines are very thick- different in any other graph in those figures. they might want to change the thickness.

We have changed the thickness of the lines to make them more consistent with other graphs in the figures.

6. Page 5 right column, line 109, "it" in " it is absolutely required..." means FANCF?
Needs clarification

Thanks for noticing this ambiguity. We have replaced "it" with "FANCF".

Referee #3:

FANCM has multiple functions in the maintenance of genome integrity. It functions as a complex with FAAP24, MFH1 and MFH2. FANCM contributes to the mono-ubiquitination of FANCI/FANCD2 (a core step in the activation of the FA pathway), and it is simultaneously a motor protein that acts during various steps of DNA repair and replication stress response.

In the beginning of the paper, it is shown in reconstituted assays that FANCM promotes FANCI/FANCD2 mono-ubiquitination (as known from before) and that FAAP24 is not required for this process. The story then follows by describing the structures of the N and C-terminal domains of FANCM, with DNA, and structural modeling that identified an interface between the N- and C-terminal domains. This part of the paper was the weakest, in my opinion, some data were overinterpreted and the advance that these data bring to the field, with respect to published literature, was not always clear. The story then follows strongly by showing that some identified DNA contact points, as well as the ATPase activity of FANCM are not necessary for FANCI/FANCD2, but are instead needed e.g. for cellular response to

DNA damage. Hence, the key functions of FANCM are separable. Overall, the manuscript contains a lot of high-quality data, but the story can be improved by improving flow and clarifying the conceptual advance.

We appreciate the reviewers assistance in improving the flow and for clarifying the conceptual advance.

Specific comments:

Fig. 1E: It is noted that FANCM promotes monoubiquitination to the point that it becomes resistant to the USP1-UAF1 deubiquitinase. While I see what the authors conclude concerning the modified upper band, there is a gradual decrease of the unmodified band (both with DANCD2 and FANCI) with increasing time of incubation in the presence of USP1-UAF1 (lanes 8-10). What is happening to the protein?

The apparent decrease you've noted in the unmodified band is not a consistent finding across our experimental replicates. After reviewing multiple repeats of this experiment, we believe this particular gel shows a minor technical variation that is not representative of the overall pattern we observe. "Source data" for this figure now includes a second repeat which does not show any difference in total FANCD2 or FANCI levels in these conditions. Such variations can occasionally occur due to slight differences in protein loading, transfer efficiency during Western blotting, or other technical factors that affect band intensity.

Also, the legend notes that statistics is shown, but the size of the graphs makes any error bars invisible. Y axis is unlabelled.

Upon review, we discovered a misplacement of the figure panel designation in the legend. The reference to "+/- standard deviation" was incorrectly associated with this panel when it should have been linked to panel 1F. We have now corrected this error in the revised figure legend.

Regarding the Y-axis labelling, we originally used "%" as shorthand due to space constraints in this comprehensive figure. However, we recognise this lacks clarity. In our revision, we have maintained the concise labelling on the axis itself but have expanded the figure legend to explicitly state that "%" refers to the percentage of ubiquitinated form relative to total protein.

Fig. 1G: Same comment on quantification as above applies to panel G. The labeling makes the experiment somewhat difficult to understand. Legend notes "complete reaction", while it is not immediately clear what is meant;

We have relabelled this from "complete reactions" to "ubiquitination reactions"

the legend should note what is quantified; also, the text refers to lanes 12-14, but no lane numbers are given.

We have added lane numbers to 1G to improve clarity.

Do the authors know that their preparation of the complex containing MHF1 and MHF2 increases motor activity of FANCM? The conclusion on the separation of function can only be made with this control.

Yes, this data is shown in Figure 5E.

Figure 2-3: Please include residue numbers on the primary structure cartoons.

Residue numbering that defines the crystal structure boundaries has now been added to 2A and 3A.

Figure 3 shows the structure of the C-terminal region of FANCM together with FAAP24. A similar complex has already been structurally analyzed in Ref32: "Structural insights into the functions of the FANCM-FAAP24 complex in DNA repair". While Ref32 is cited, it was in the general context that FAAP24 is needed for FANCM function. The advance of the new structure should be made more clear with respect to the previous data.

We have introduced a more detailed description of the significant advance over our previous structure, to more clearly point out the 1) new DNA binding site identified, 2) the larger overall DNA binding footprint and 3) the lack of engagement with the junctions DNA.

Figure 4: While Figure 2 and Figure 3 show crystal structures of FANCM's N-terminal and C-terminal domains with DNA in isolation, Figure 4 combines these data with AF3 modeling to identify potential interface sites that bring the N- and C-terminal domains together. A mutation is analyzed, which is predicted to disrupt the interface (i.e. ternary structure), and which results in defects in cell survival and foci experiments. While the model is most likely correct, I do not see how these data support the conclusion: "These findings reveal FANCM's sophisticated DNA damage recognition system, where the motor domain identifies branch points while the C-terminal domains provide additional grip and positioning."

I agree that the word "reveal" suggests that we have actually seen the association, which we haven't, so we changed the word "reveal" to "suggest".

Along the same lines, it is noted "The discovery that FANCM's domains physically interact suggested they might functionally cooperate in migrating these branched DNA structures." It is known from the previous literature (e.g. Ref32) that the C-terminal domain targets FANCM to chromatin, and as the motor domain is in the N-terminus, it is not clear how the data conceptually advance the field.

Ref32 did include some experiments that showed the C-terminus is required for targeting FANCM to DNA, but it hasn't previously been shown that the two regions of FANCM physically interact or cooperate in branch migration. This is a connecting sentence, not a strong conclusion – and is necessary to justify the experiments performed in Figure 5. Because previous gel based assays and FANCM purification methods were inferior to the new reported assay, the substantial differences in efficiency of the enzyme with/without MHF and/or FAAP24 were missed.

Could the authors demonstrate whether the discovered separation of function mutations differently impact the cellular functions of FANCM? The discovery that the ATPase and DNA binding mutants are not needed for FANCI/FANCD2 mono-ubi but are required for branch migration is exciting, but I feel that the story somehow falls short to demonstrate how these two (newly discovered) distinct processes impact on the processes in which FANCM promotes genome stability. While this may be above the scope of a revision, I feel that this would really bring the field forward.

We appreciate your enthusiasm about the discovery that ATPase and DNA binding mutants are not required for FANCI/FANCD2 mono-ubiquitination but are essential for branch migration.

Our current work has demonstrated that the ATPase activity of FANCM can be uncoupled from its FA core complex stimulation activity specifically in terms of DNA damage sensitivity. We fully agree that determining how these distinct processes impact FANCM's role in genome stability would significantly advance the field.

The FANCM ATPase domain participates in multiple cellular functions, and pinpointing which specific role is responsible for the continued sensitivity of FANCM ATPase mutants to crosslinking agents would require extensive additional investigations. These would likely include creating and characterising a comprehensive panel of separation-of-function mutants, followed by detailed cellular and biochemical analyses across various DNA damage response pathways.

While we believe this represents an important direction for future research, such comprehensive functional dissection would require substantial additional resources and time beyond what's feasible for the current revision timeframe. We have therefore focused on solidly establishing the separation of these functions in this manuscript, providing a foundation for these more detailed mechanistic studies moving forward.

We hope to address these fascinating questions in follow-up work and appreciate your thoughtful suggestion which will help shape our future research directions.

Prof. Andrew J Deans
St Vincent's Institute
Genome Stability Unit
9 Princes St
Fitzroy, VIC 3065
Australia

12th May 2025

Re: EMBOJ-2025-120753R
Structural basis of Fanconi Anemia pathway activation by FANCM

Dear Andrew,

Thank you for submitting your revised manuscript to The EMBO Journal. One of the original referees has now assessed it once more, and I am happy to say was fully satisfied with the revisions. After incorporation of the following remaining editorial issues, we should therefore be able to proceed with formal acceptance of the study:

- Please carefully go through the reference list and make sure that each reference is complete with citation year, volume, and page/locator numbers.
- In the Data Availability section, please add an additional URL directly linking to the modelarchive database (which is currently only included in a different subsection of the methods).
- Please remove all Appendix figure legends from the main manuscript file, they should only appear in the Appendix itself.
- Please make sure to adjust the figure scale bar indications in Figs. 4, 6, and 7 to make them consistent with the provided images.
- For Figure EV1B, where $n=2$ in some of the data points, thereby not justifying statistical analyses such as generation of error bars: I would suggest to replot the complete panel without error bars, for each concentration simply plotting all 2 or 3 individual data points plus a line indicating the mean value. Furthermore, for clarity it may be helpful to label the x-axis with the respective salt concentrations, which are currently only mentioned in the legend.

I am returning the manuscript to you for a final round of minor revision, solely to allow you to make these modifications and upload the revised files. Once we will have received them, we should be ready to swiftly proceed with formal acceptance and production of the manuscript.

With kind regards,

Hartmut

- 2) Each figure legend must specify
- size of the scale bars that are mandatory for all micrograph panels
 - the statistical test used to generate error bars and P-values
 - the type error bars (e.g., S.E.M., S.D.)

- the number (n) and nature (biological or technical replicate) of independent experiments underlying each data point
- Figures may not include error bars for experiments with $n < 3$; scatter plots showing individual data points should be used instead.

9) To facilitate reproducibility and cross-laboratory adoption of methodologies, please structure the Materials & Methods section as outlined in our guide to authors, including a completed Reagents and Tools Table that can be downloaded from our author guidelines as well (<https://www.embopress.org/page/journal/14602075/authorguide#structuredmethods>).

10) Digital image enhancement is acceptable practice, as long as it accurately represents the original data and conforms to community standards. If a figure has been subjected to significant electronic manipulation, this must be clearly noted in the figure legend and/or the 'Materials and Methods' section. The editors reserve the right to request original versions of figures and the original images that were used to assemble the figure. Finally, we generally encourage uploading of numerical as well as gel/blot image source data; for details see: embopress.org/page/journal/14602075/authorguide#sourcedata

In the interest of ensuring the conceptual advance provided by the work, we recommend submitting a revision within 3 months (10th Aug 2025). Please discuss the revision progress ahead of this time with the editor if you require more time to complete the revisions. Use the link below to submit your revision:

Link Not Available

Referee #3:

I am happy to support the revised manuscript for publication. The authors have answered all my comments, and I agree with everything. Congratulations on the nice work